# Nanoconfinement steers nonradical pathway transition in single atom fenton-like catalysis for improving oxidant utilization

Yan Meng [1,2,5], Yu-Qin Liu [1,5], Chao Wang [3,5], Yang Si[4], Yun-Jie Wang [1,2], Wen-Qi Xia [1,2], Tian Liu[2], Xu Cao [1], Zhi-Yan Guo [1,2] ✉, Jie-Jie Chen [1] & Wen-Wei Li [1,2] ✉

The introduction of single-atom catalysts (SACs) into Fenton-like oxidation promises ultrafast water pollutant elimination, but the limited access to pollutants and oxidant by surface catalytic sites and the intensive oxidant consumption still severely restrict the decontamination performance. While nanoconfinement of SACs allows drastically enhanced decontamination reaction kinetics, the detailed regulatory mechanisms remain elusive. Here, we unveil that, apart from local enrichment of reactants, the catalytic pathway shift is also an important cause for the reactivity enhancement of nanoconfined SACs. The surface electronic structure of cobalt site is altered by confining it within the nanopores of mesostructured silica particles, which triggers a fundamental transition from singlet oxygen to electron transfer pathway for 4-chlorophenol oxidation. The changed pathway and accelerated interfacial mass transfer render the nanoconfined system up to 34.7-fold higher pollutant degradation rate and drastically raised peroxymonosulfate utilization efficiency (from 61.8% to 96.6%) relative to the unconfined control. It also demonstrates superior reactivity for the degradation of other electron-rich phenolic compounds, good environment robustness, and high stability for treating real lake water. Our findings deepen the knowledge of nanoconfined catalysis and may inspire innovations in low-carbon water purification technologies and other heterogeneous catalytic applications.

Water pollution due to inadequate removal of recalcitrant contaminants remains a pervasive global challenge threatening sustainable water supply and human health[1,2]. Thus, economically affordable and efficient water decontamination technologies are highly desired[3]. Heterogeneous Fenton-like oxidation technologies hold great promise in this line[4,5], where reactive species such as hydroxyl radicals (•OH), sulfate radicals (SO$_4^{•-}$), and nonradical species[6,7] can be generated for efficiently degrading organic pollutants. Especially, there are growing interest in preferentially generating nonradical species including singlet oxygen ($^1O_2$), high-valent metals, and surface-bound oxidative complexes, which allow selective oxidation of electron-rich pollutants and hence more efficient utilization of oxidants than in radical reaction systems[8–10]. Among the heterogeneous catalysts for nonradical Fenton-like reactions[11,12], single-atom catalysts (SACs) are of particular interest. Being highly reactive and structurally tunable, SACs can be elaborated for further raised reaction kinetics and less oxidant input

[1]CAS Key Laboratory of Urban Pollutant Conversion, Department of Environmental Science and Engineering, University of Science & Technology of China, Hefei, China. [2]Sustainable Energy and Environmental Materials Innovation Center, Suzhou Institute for Advanced Research, University of Science & Technology of China, Suzhou, China. [3]National Synchrotron Radiation Laboratory, University of Science & Technology of China, Hefei, China. [4]Kunming Institute of Physics, Kunming, China. [5]These authors contributed equally: Yan Meng, Yu-Qin Liu, Chao Wang. ✉e-mail: gzy2018@ustc.edu.cn; wwli@ustc.edu.cn

than the metal-based nanomaterials[13,14]. Nevertheless, the overall decontamination performance is still limited by the inadequate reactant access to the surface active sites and the short diffusion range of the generated active species, dominated by $^1O_2$ in the existing SAC-based Fenton-like catalytic systems[15,16]. These challenges can not be fully addressed by common material optimization strategies such as heteroatom doping and vacancy introduction[17,18].

Recently, nanoconfinement has emerged as another promising strategy for catalyst engineering. Confining the SAC active sites within a nano-structured space was found to drastically raise the local concentration of reactive species and pollutants[19,20], rendering the Fenton-like catalytic system 2–3 order-of-magnitude higher reactivity for pollutant degradation than the unconfined analog[21,22]. In addition, nanoconfinement could also alter the physicochemical properties of metal-based nanoparticles, thereby affecting the thermodynamics of surface-reactant interactions and even causing catalytic pathway change[23,24]. For example, confining the Fe(III)-anchored metal-organic framework within the porous graphene aerogel triggered a fundamental transition in the Fenton-like catalytic pathway for phenol oxidation from ring-opening degradation to oligomerization route[25]. Such changes imply a possibility for addressing the activity, selectivity, and interfacial mass transfer issues of catalysts simultaneously. Nevertheless, it remains unclear whether and how the surface catalytic behaviors of SACs for pollutant degradation would be affected under nanoconfinement conditions.

Here, we demonstrate that, apart from strengthening the local enrichment and diffusion of reactants, nanoconfinement could also fundamentally alter the coordination configuration and electronic structure of the active site of SACs to trigger catalytic pathway change. In this proof-of-concept study, we constructed a model catalytic system by confining Co single atoms within the nano-sized pores of carbon-coated mesoporous silica particles (MSi, consisting of a spherical silica core and a mesoporous shell), where the nitrogen-doped carbon layer provides an abundant nanoconfined surface for anchoring Co atoms. The nanopore sizes of the MSi shell were fine-tuned to adjust the nanoconfinement degree. For comparison, the carbon-coated nonporous silica particles (Si) were adopted as the unconfined control. By using PMS as the oxidant and 4-chlorophenol (4-CP) as the model pollutant, we unveil a fundamental transition of the pollutant degradation pathway from $^1O_2$ to electron transfer process (ETP) under nanoconfinement. Such pathway change, together with the accelerated interfacial mass transfer under nanoconfinement, led to 34.7-fold increased decontamination kinetics and drastically raised PMS utilization efficiency (PUE, from 61.8% to 96.6%) in Fenton-like catalysis. The mechanisms underpinning the catalytic pathway transition were elucidated by experimental study and density functional theory (DFT) calculations. In addition, the nanoconfined catalytic system with CoNC-MSi exhibited superior salinity resistance and environmental robustness, and maintained high stability for real lake water treatment during long-time operation in both continuous-flow packed-bed and catalytic membrane reactors.

## Results
### Characteristics of nanoconfined Co SACs
Two nanoconfined Co-SACs and one unconfined control, with similar particle sizes, were fabricated following the procedures illustrated in Supplementary Fig. 1. As shown by the scanning electron microscopic (SEM) and transmission electron microscopic (TEM) images, the nanoconfined catalysts with thicker shells exhibited relatively larger particle sizes (~400 nm) than the CoNC-Si (~310 nm) (Fig. 1a, b and Supplementary Fig. 2). The $N_2$ adsorption-desorption isotherms and the pore size distribution curves reveal smaller average pore sizes of CoNC-MSi1 (7.7 nm) than CoNC-MSi2 (12.8 nm) (Fig. 1c and Supplementary Table 1), in line with the more densely-distributed nanopores on the shell of CoNC-MSi1 (Fig. 1b and Supplementary Fig. 2). In addition, the porous

shell rendered the CoNC-MSi1 6.8-fold larger specific surface area than the nonporous CoNC-Si (Supplementary Table 1).

The large surfaces of the nanoconfined catalysts afford abundant anchoring sites for Co single atoms. Atomically dispersed bright spots are clearly shown by the aberration-corrected high-angle annular dark-field scanning transmission electron microscopic (AC-HAADF-STEM) images (Fig. 1e). In addition, the energy-dispersive spectroscopy (EDS) elemental mapping (Fig. 1d and Supplementary Fig. 3) reveals homogeneously distributed Co, N, and C elements across the silica architecture of all the catalyts[26], confirming the presence of Co single atoms on the N-doped carbon surface layer[27] (Fig. 1f). The Co loading was validated by the inductively coupled plasma (ICP) measurement results, which show Co contents of CoNC-MSi1 (0.07 wt%) > CoNC-MSi2 (0.05 wt%) > CoNC-Si (0.01 wt%) (Supplementary Table 2), consistent with the AC-HAADF-STEM observation of much less bright spots for the CoNC-Si (Fig. 1e). The limited Co atoms loading explains the weaker EDS signal of Co than the other elements (Fig. 1d). No obvious signal of Co-Co bond was detected by the X-ray diffraction (XRD), Raman, and FT-IR spectra for all the catalysts, confirming that the Co element was present predominantly in the form of single atoms[28,29] (Supplementary Figs. 4–6). It is noteworthy that the bright spots on CoNC-MSi1 and CoNC-MSi2 were mainly distributed in the nanopore region (Fig. 1e), indicating a nanoconfinement state of the Co atoms that were in stark contrast with the unconfined ones in CoNC-Si.

Apart from increased Co SACs loading, the coordination configuration and valence states of the Co atoms were also drastically altered under nanoconfinement. The N 1$s$ X-ray photoelectron spectra (XPS) show distinct signals of Co-N bond (a typical feature of N-coordinated Co SAC[30]), whose peak intensity was increased under nanoconfinement as a result of more Co atoms loading (Fig. 2a and Supplementary Tables 3 and 4). The Co $K$-edge X-ray absorption near-edge structure (XANES) and extended X-ray absorption fine structure (EXAFS) spectra reveal more detailed valence and coordination information of the Co sites. The Co valences of the catalysts were all between 0 and 2 and decreased by the order of CoNC-MSi1 > CoNC-MSi2 > CoNC-Si (Fig. 2b). In addition, the Fourier-transformed (FT) $k^3$-weighted EXAFS spectra of all the catalyst show a distinct peak at 1.4–1.6 Å, corresponding to the Co-N first-shell coordination[31] (Fig. 2c). A wavelet transform (WT) peak at 5 Å$^{-1}$, ascribed to the Co-N bond, was also identified[32] (Fig. 2d–h). According to the EXAFS fitting results (Supplementary Table 5 and Fig. 7), the Co-$N_4$ coordination configuration dominated in the nanoconfined Co SAC, while Co-$N_3$ prevailed in the unconfined control[33]. Notably, the Co-Co signal was also detected in the CoNC-Si by the EXAFS and Co 2$p$ XPS spectra (Fig. 2c and Supplementary Fig. 8), indicating a slight agglomeration of the unconfined Co atoms although it counted for only ~3 wt % of the total Co contents[34]. Overall, these results suggest an increased N coordination number of the Co atoms under nanoconfinement (Fig. 2i).

### Pollutant degradation and PMS utilization performances
Attributed to the different Co loading and coordination configurations, the nanoconfined Co SACs exhibited drastically different Fenton-like oxidation behaviors from those of the unconfined analog. A complete removal of 4-CP within 2 min was achieved by the CoNC-MSi1 in the presence of PMS (against only ~20% 4-CP removal by the CoNC-Si) (Fig. 3a and Supplementary Figs. 9 and 10), corresponding to a 34.7-fold higher degradation rate constant for the CoNC-MSi1 (Fig. 3b). Further normalizing the kinetic constant to specific surface area ($k_{SA}$, reflecting the specific activity) and to the Co loading amount ($k_{per-site}$, reflecting the intrinsic reactivity of Co atoms)[35] still yield 10.3-fold and 5.1-fold higher values for the CoNC-MSi1 over CoNC-Si (Fig. 3c and Supplementary Fig. 11). Consistent results were obtained for the estimated turnover frequency (TOF; 4-CP degradation rate at per Co atom basis)[36] (Supplementary Fig. 12). All these evidences strongly

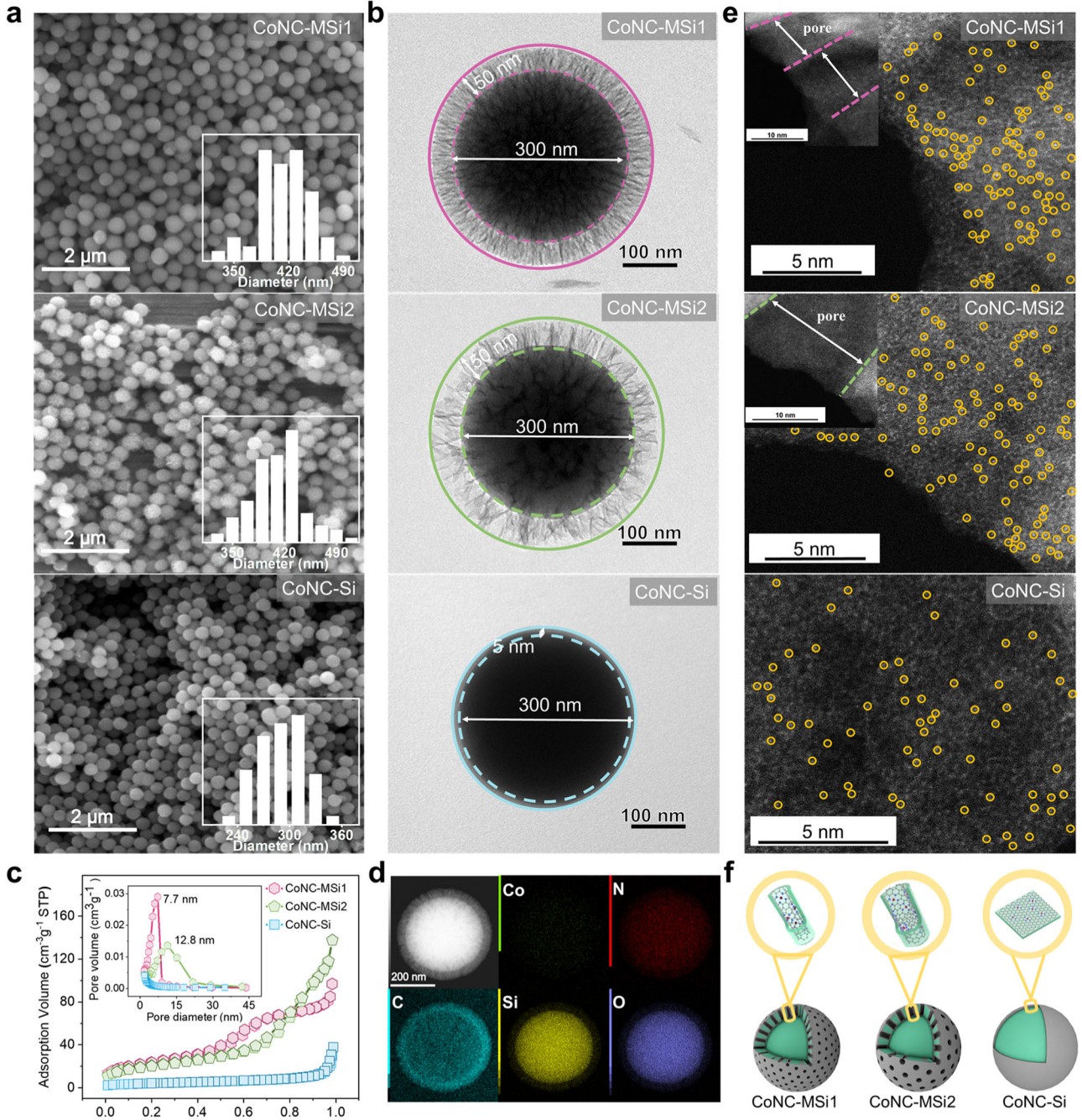

**Fig. 1 | Morphological and structural properties of nanoconfined CoNC-MSi and unconfined CoNC-Si catalysts. a** SEM image and the particle size distribution. **b** TEM images. **c** $N_2$ adsorption-desorption isotherms and the pore size distribution curves of the catalysts. **d** EDS mapping of CoNC-MSi1 catalyst. **e** AC-HAADF-STEM images showing the distribution of single-atom Co (marked with the yellow circles, SACs were mainly located inside the pores of the catalyst shell). **f** Schematic diagram of the catalyst spatial structure.

support a profoundly raised intrinsic reactivity of the Co atoms under nanoconfinement. Notably, the 4-CP degradation reactivity of CoNC-MSi2 was also enhanced relative to CoNC-Si but to a less extent (Fig. 3a–c), indicating a size-dependent nanoconfinement effect[37].

Here, the reactivity of CoNC-Si should be mainly ascribed to its Co single atoms because its decontamination performance remained almost unchanged after acid treatment to remove the minor Co nanoclusters (Supplementary Fig. 13). In addition, the Co-free controls (including NC-MSi1, C-MSi1, and MSi1) and the non-SAC control (including homogeneous $Co^{2+}$, $Co_2O_3$, and $Co_3O_4$), all exhibited weak or negligible reactivity for 4-CP degradation (Fig. 3a and

Supplementary Fig. 14). Therefore, the drastically raised reactivity of CoNC-MSi1 relative to CoNC-Si should be ascribed to both increased loading of Co single atoms and raised intrinsic activity of the Co active sites under nanoconfinement. Notably, our CoNC-MSi1/PMS system exhibited remarkable pollutant degradation activity ($k_{SA} = 0.004\ \mathrm{g\,m^{-2}\,min^{-1}}$), outperforming almost all the reported nonradical-dominated heterogeneous catalytic systems (Fig. 3c and Supplementary Table 6).

In addition to the raised activity, the nanoconfined Co SAC also enabled more efficient PMS utilization. Here, the PMS utilization efficiency (PUE) is defined as the ratio of the equivalent amount of PMS (the sole oxidant showing in Supplementary Fig. 15) consumed for

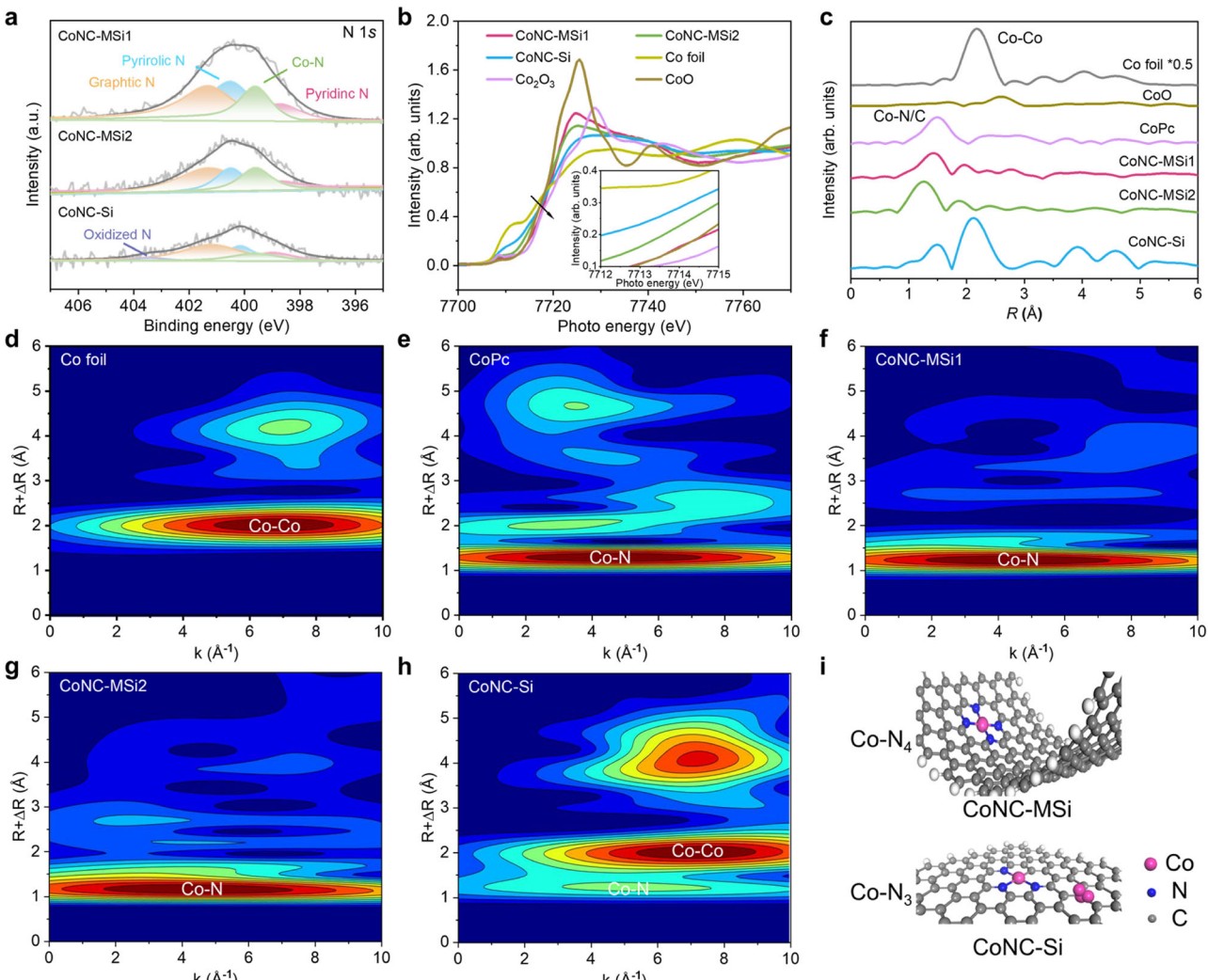

**Fig. 2 | Coordination geometry and chemical states of the catalysts. a** N 1*s* XPS spectra of different Co SACs. **b** Normalized Co *K*-edge XANES spectra of the Co SACs and reference samples (Co foil, CoO, and Co₂O₃). **c** Co *K*-edge FT-EXAFS spectra of the Co SACs and reference samples (Co foil, CoO, and CoPc). **d–h** WT-EXAFS plots of different catalysts. **i** Schematic diagram of coordination configurations of the Co SACs.

4-CP mineralization to the amount of decomposed PMS within a given time frame. The PUE value was estimated according to the TOC change and PMS consumption in the catalytic systems[38] (Supplementary Fig. 16). Strikingly, the CoNC-MSi1/PMS system achieved an extraordinary PUE of 96.6% for 4-CP degradation, surpassing those of the CoNC-Si/PMS system (61.8%) and most reported nonradical catalytic systems (Fig. 3b, c and Supplementary Table 6).

**Catalytic pathways transition triggered by nanoconfinement**
The raised reactivity and PUE of the Co SAC under nanoconfinement were associated with the raised local concentration and improved diffusion of reactants over the catalyst surface. According to the molecular dynamic (MD) simulation results, the CoNC-MSi1 enables a 2.1-fold raised PMS concentration and 4.3-fold raised 4-CP concentration at the proximity of the pore wall relative to those in the bulk liquid, indicating an obvious surface enrichment of reactants. In contrast, the reactant enrichment effect was much weaker for the CoNC-MSi2, while almost uniform reactant concentration distribution was observed for the CoNC-Si (Fig. 3d and Supplementary Fig. 17), indicating a nanoconfinement-induced reactant enrichment. In addition, the mean square displacement (MSD) curves (Fig. 3e) suggest a much faster rate of reactant diffusion for the CoNC-MSi1[39], thus contributing considerably to its superior reactivity. Notably, the CoNC-MSi1 showed tenfold higher

specific activity than the CoNC-Si but only up to 4.3-fold raised local reactant concentration, indicating that some other factors might also account for the performance improvement under nanoconfinement.

Notably, besides the raised catalytic activity and reactant enrichment, the catalytic pathway of the Co SAC also underwent fundamental change under nanoconfinement. According to the results of reactive oxygen species (ROS) quenching experiments[40] (Fig. 3f and Supplementary Figs. 18 and 19), the pollutant degradation in the CoNC-Si/PMS system was severely suppressed by adding furfuryl alcohol (FFA) as a singlet oxygen (¹O₂) scavenger[41]. The D₂O experiments and electron spin resonance (ESR) analysis with a TEMP probe also supported the abundant generation of ¹O₂ in this system[42] (Fig. 3h and Supplementary Fig. 18). In contrast, no radicals and only weak ¹O₂ signal were detected in the nanoconfined catalytic systems, suggesting the prevalence of other nonradical pathways (Fig. 3g, h and Supplementary Figs. 18 and 19). Further quantification of the ¹O₂ generated in the different catalytic systems with 1,3-diphenylisobenzofuran (DPBF) probe[43,44] showed different proportions of ¹O₂ pathway for the CoNC-MSi1 (33%), CoNC-MSi2 (47%) and CoNC-Si (98%) (Fig. 3i). All these evidences strongly prove a transition from ¹O₂ to other nonradical pathways of the Co SAC triggered by nanoconfinement.

Apart from ¹O₂, high-valent metal-oxo and surface-bound PMS complexes (PMS*) were also commonly found in nonradical

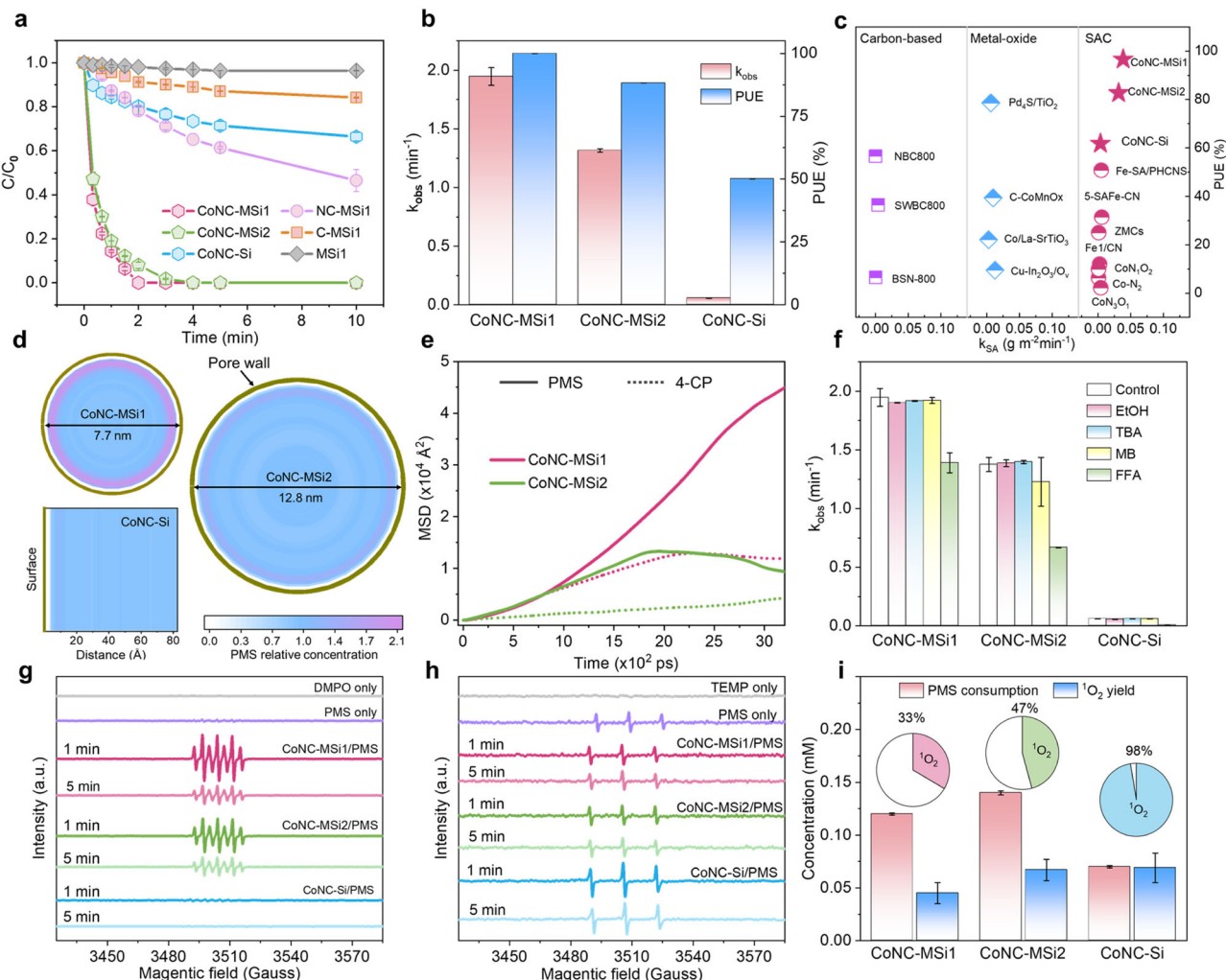

**Fig. 3 | Nanoconfinement effects analysis on Fenton-like catalytic activity, mass transfer, and pathway. a** 4-CP degradation performances. **b** Corresponding kinetic constant ($k_{obs}$) and PUE of different catalyst/PMS systems. Error bars represent the standard deviation, obtained by repeating the experiment twice. **c** Comparison with other catalysts in PUE and specific activity for pollutant oxidation. **d** Concentration distribution of PMS obtained by molecular dynamic simulations. **e** MSD results showing reactant diffusion kinetics for different catalysts. **f** Effects of different scavengers on 4-CP degradation kinetics and the fraction of 4-CP removal contributed by $^1O_2$ pathway. EtOH, TBA, MB, and FFA were used to detect $\cdot OH$, $SO_4^{\cdot-}$, surface-bound radicals, and $^1O_2$, respectively. Error bars represent the standard deviation, obtained by repeating the experiment twice. **g, h** ESR spectra. **i** The amount of PMS consumed for 4-CP degradation and $^1O_2$ generation (equivalent amount, estimated according to the DPBF conversion, and the pie charts indicate the estimated ratio of $^1O_2$ pathway in the three systems). Error bars represent the standard deviation, obtained by repeating the experiment twice. Reaction condition: [catalyst] = 0.25 g L$^{-1}$, [PMS] = 0.4 mM, [4-CP] = 0.1 mM, [EtOH] = [TBA] = 400 mM, [MB] = 0.1 mM, [FFA] = 4 mM, [DPBF] = 0.4 mM.

Fenton-like catalyis[45,46], thus their possible generation by the nanoconfined Co SAC was also examined. However, no signal of high-valent Co-oxo species (Co$^{IV}$) was detected by the dimethyl sulfoxide (DMSO) quenching experiment and by the methyl phenyl sulfoxide (PMSO) probe[47] (Supplementary Figs. 18 and 20), thus excluding the high-valent metal pathway. Meanwhile, an obvious pollutant-dependent PMS consumption behavior of CoNC-MSi1 and CoNC-MSi2 was observed, implying the predominance of PMS*-based ETP pathway[48] (Supplementary Fig. 21). Consistently, the in-situ Raman spectra of the catalyst identify a distinct peak that occurred at ~835 cm$^{-1}$ upon PMS addition but disappeared after subsequent 4-CP addition, corresponding to the processes of PMS* formation and consumption[40] (Fig. 4a). In addition, the chronoamperometric current and open-circuit potentials (OCP) of the nanoconfined catalysts showed sharp changes in response to PMS and 4-CP addition (also typical of ETP pathway[49]) (Fig. 4b, c), but no discernable changes were identified for the CoNC-Si. All these evidences confirm that the catalytic pathway of Co SAC shifted from $^1O_2$ pathway to ETP-dominated pathway under

nanoconfinement, thus also contribute considerably to the raised activity and PUE of CoNC-MSi1.

In the ETP decontamination pathway, the pollutant may be directly oxidized by the PMS* or by the catalyst as a conductive bridge (i.e., mediated ETP)[12], depending on the intensity of binding and electronic interaction between the pollutant/ oxidant and the catalyst[50]. To elucidate the specific catalyst-pollutant-PMS interactions, a galvanic cell was adopted, where the two electrolyte chambers (containing 4-CP and PMS, respectively) were separated by a catalyst-loaded carbon flake. The system with CoNC-MSi1-loaded electrode exhibited the largest current decrease upon PMS addition, accompanied by synchronous 4-CP degradation in the separated chamber (Fig. 4d, e and Supplementary Fig. 22). The result of 100% 4-CP degradation after 6-hour reaction in the galvanic cell provided strong evidence of catalyst-mediated electron transfer from 4-CP to surface PMS* complex. In addition, the CoNC-MSi1 group showed raised open-circuit potential with reactant concentration[51] (Fig. 4f and Supplementary Fig. 23), further validating a mediated ETP pathway with PMS* generation. This

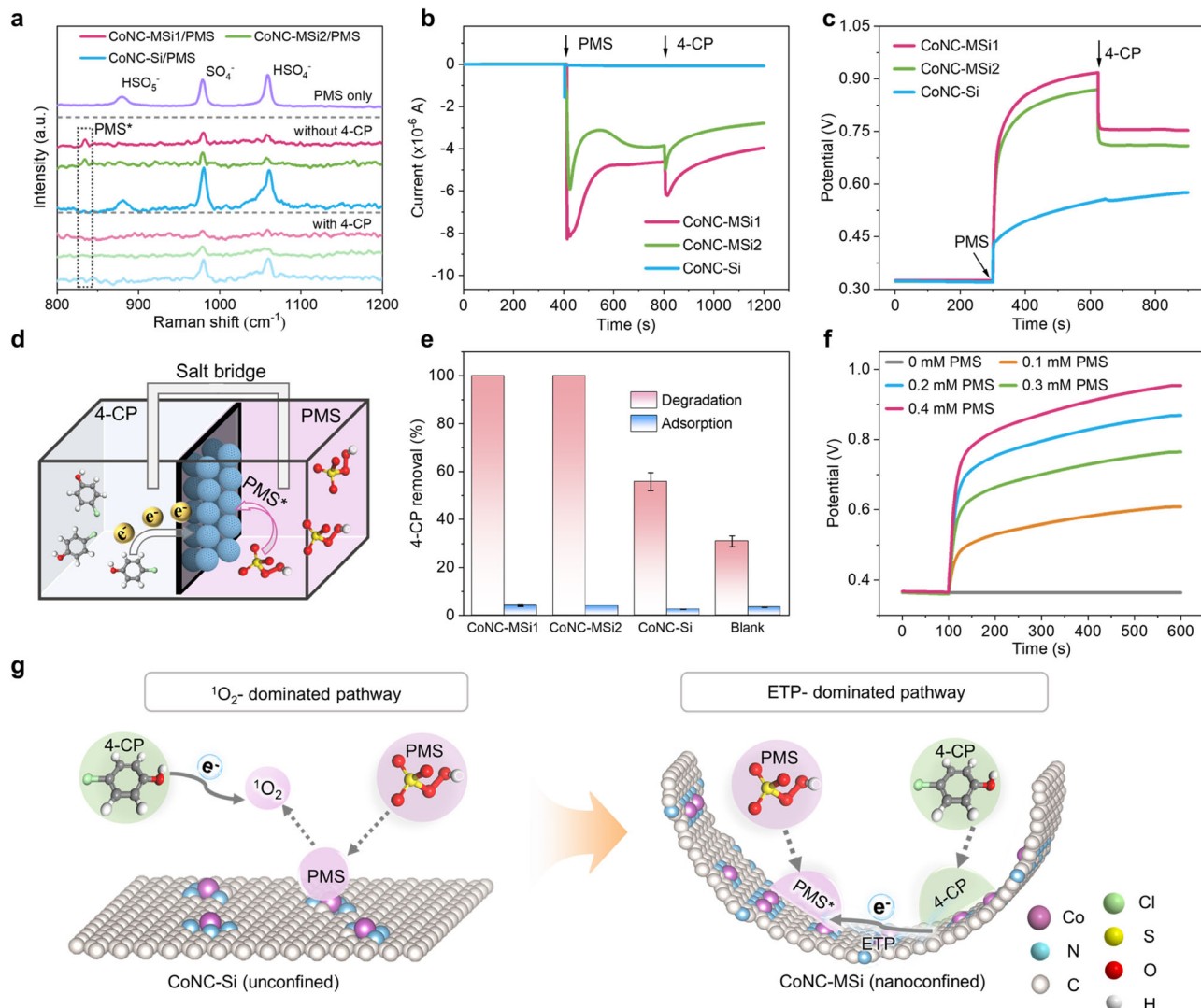

**Fig. 4 | Mechanistic investigation of nanoconfined Fenton-like catalytic system for decontamination. a** In situ Raman spectra. **b** $I$–$t$ (current-time) curves and **c** open-circuit potentials (OCP) changes upon sequential addition of PMS and 4-CP. **d** Schematic diagram, and **e** 4-CP removal performances of the galvanic cell during 6-h operation. The catalysts were loaded on both sides of the carbon separator between the two chambers. Error bars represent the standard deviation, obtained by repeating the experiment twice. **f** OCP of CoNC-MSi1-modified electrode with different PMS concentrations. **g** Schematics of the different catalytic pathways for the unconfined and nanoconfined Co SACs. Reaction condition for **d**–**e**: [catalyst] = 0.5 g L$^{-1}$, [PMS] = 0.8 mM, [4-CP] = 0.05 mM.

---

pathway was enabled by the inner-sphere binding of both the pollutant and PMS by the nanoconfined catalyst, as evidenced by its high resistance to NaClO$_4$ interference (Supplementary Fig. 24)[52], as well as its high conductivity (electric resistance ~480 Ω, graphitic degree $I_D/I_G = 0.86$)[53] (Supplementary Figs. 5 and 25).

Therefore, a $^1O_2$-to-ETP pathway transition in Fenton-like catalysis of Co SAC was triggered by nanoconfinement. Specifically, in the unconfined system, a considerable fraction of the surface generated $^1O_2$, due to lack of sufficient contact with pollutants, was self-decomposed during diffusion into the bulk solution[54], resulting in low PUE. In contrast, the nanoconfined system not only enabled raised local concentrations of reactants to accelerate PMS activation and pollutant oxidation, but also provide long-lasting PMS* for effective degradation of target pollutants (Fig. 4g), thereby contributing considerably to its high decontamination activity and PUE.

## Mechanisms of nanoconfinement-induced catalytic pathway change

It is interesting to know what material property changes, triggered by nanoconfinement, have driven the catalytic pathway shift in the Co SAC catalytic system. No straightforward correlation was found between the Co content and $^1O_2$/ ETP pathway, suggesting that the catalytic pathway change under nanoconfinement was not ascribed to increased Co loading (Supplementary Fig. 26). Previous studies suggest that nanoconfinement can fundamentally alter the coordination environment and electronic properties of the confined molecules, which in turn affect the interfacial reaction kinetics and thermodynamics[44,55]. Here, discrepant N coordination numbers were identified for the nanoconfined and unconfined Co SACs (Fig. 2i and Supplementary Table 5). According to the DFT calculations, changing from Co-N$_3$ to Co-N$_4$ configuration for the unconfined structures marginally increases the electron transfer number (−0.70 e vs. −0.77 e) and PMS adsorption ($E_{ads}$ = −2.56 eV vs. −2.57 eV) (Supplementary Figs. 27 and 28 and Supplementary Table 7), suggesting that N coordination number alone does not govern the catalytic pathway. Notably, the increased N coordination number under nanoconfinement renders the Co sites more empty $3d$ orbitals, as evidenced by raised Co valence (Fig. 2b), for hybridization with O $2p$ orbitals of PMS molecules[46,56]. Consequently, the nanoconfined Co SACs can provide 9% increased PMS binding energy and 16.7% raised interfacial charge transfer

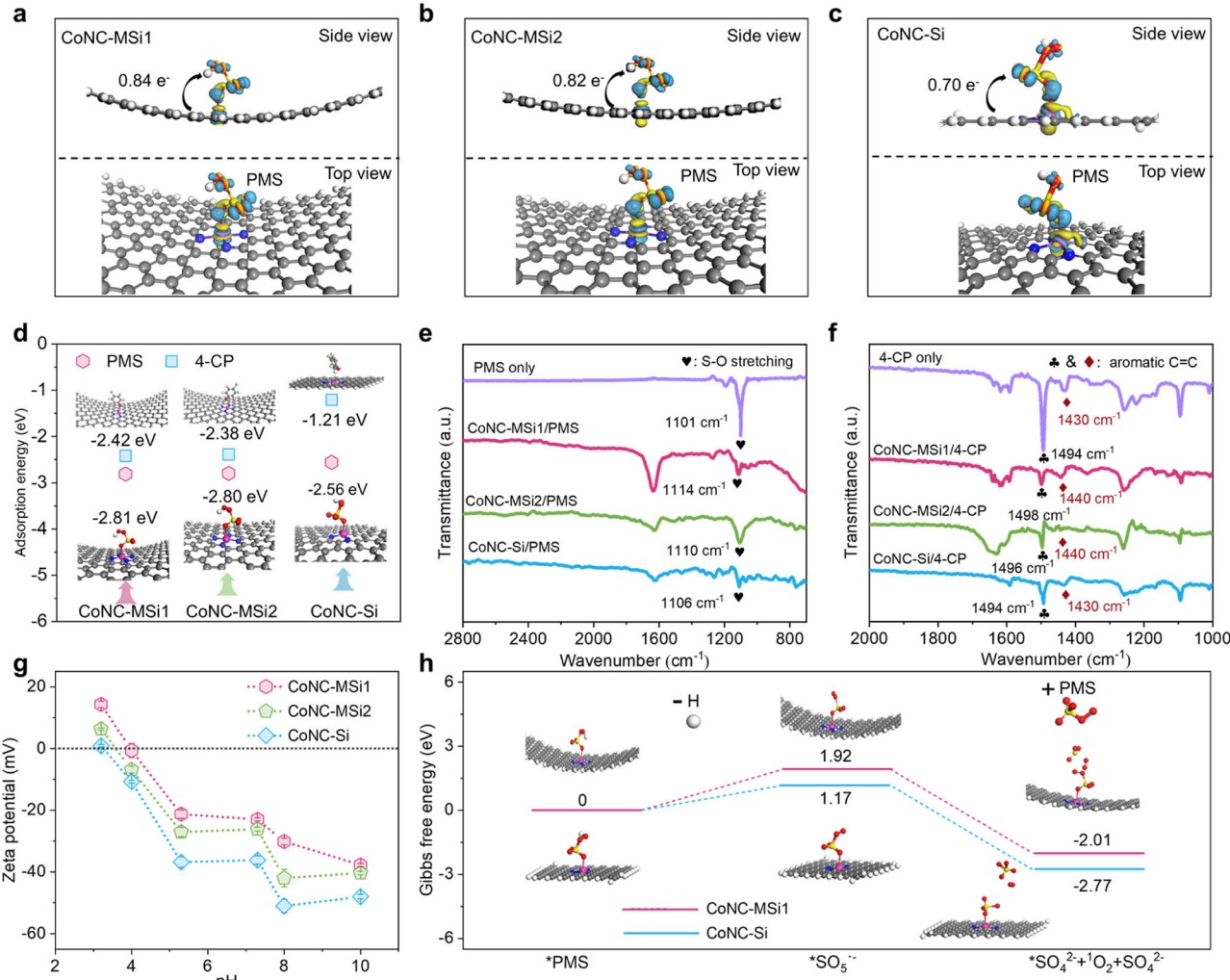

**Fig. 5 | Molecular mechanisms of nanoconfinement-induced PMS activation and pollutant removal identified by interfacial interaction. a–c** Charge density difference of PMS adsorption on different catalysts (the light yellow and blue color denote the electron depletion and accumulation regions, respectively). **d** Models and affinity of PMS and 4-CP binding on different catalysts. **e, f** In situ FT-IR spectra of PMS and 4-CP with different catalysts. **g** Zeta potential of the catalysts. Error bars represent the standard deviation, obtained by repeating the experiment twice. **h** Pathway of $^1O_2$ generation over different catalysts.

number than the unconfined control to facilitate the PMS* formation (Fig. 5a–d and Supplementary Table 7). These results reveal an enhanced ETP pathway for nanoconfined Co SACs with increased N coordination and underline the potential of other nanoconfinement effects in redirecting the catalytic pathways.

To decipher the other key factors governing the catalytic pathway change under nanoconfinement, we further explored into the catalyst surface chemistry and catalyst-PMS interactions. The FT-IR spectra show an obvious red-shift in the feature peaks of both PMS and 4-CP after mixing with the CoNC-MSi1 (Fig. 5e, f), confirming their strong surface binding[57]. However, the same was not observed for CoNC-Si. Additionally, the CoNC-MSi1 exhibited more surface -OH groups (Supplementary Fig. 29) with drastically raised point of zero charge (PZC)[58] than the CoNC-Si (Fig. 5g), which also favor PMS binding via -OH substitution[59]. The strengthened binding and raised energy barrier for deprotonation of PMS under nanoconfinement (Fig. 5h) thus drives a preferentially PMS-to-PMS* conversion process and according efficient 4-CP oxidation via a mediated ETP pathway (Fig. 4g). In contrast, due to the weak interaction between PMS and the unconfined Co in the CoNC-Si, $SO_5^{\cdot-}$ was abundantly generated and spontaneously converted into $^1O_2$ ($2SO_5^- + 2e^- \rightarrow {}^1O_2 + 2SO_4^{2-}$, Gibbs free energy change

($\Delta G$)= -2.77 eV), thus rendering a $^1O_2$-dominated pathway for 4-CP degradation[60] (Fig. 5h).

Notably, the nanoconfinement effects on catalytic pathway and activities were size-dependent. Compared with the CoNC-MSi1, the CoNC-MSi2 with larger nanopore sizes showed lower average valence of Co (Fig. 2b), less surface -OH groups (Supplementary Fig. 29) and slightly weaker binding affinity for PMS and 4-CP (Fig. 5d–f). Consequently, a less fraction of ETP pathway, lower 4-CP degradation reactivity and lower PUE (82.7% vs. 96.6%) was achieved in the CoNC-MSi2/PMS than the CoNC-MSi1/PMS system (Fig. 3b). These results are consistent with the previous reports that the nanoconfinement effects typically became more prominent with reduced size of the confined space[61,62]. However, the size-dependent nanoconfinement effect may become more complicated when it comes to a level of several nanometers, because of the drastically changed solution chemistry and interfacial molecule interactions[21,63]. Here, according to our DFT calculation, further downsizing the catalyst nanopore from 7.7 nm to 2.7 nm results in weakened binding of reactants and less change transfer numbers instead (Supplementary Fig. 30 and Table 7). Overall, an optimal value of nanopore size might exist for maximizing the catalytic performance of nanoconfined Co SACs, although the detailed

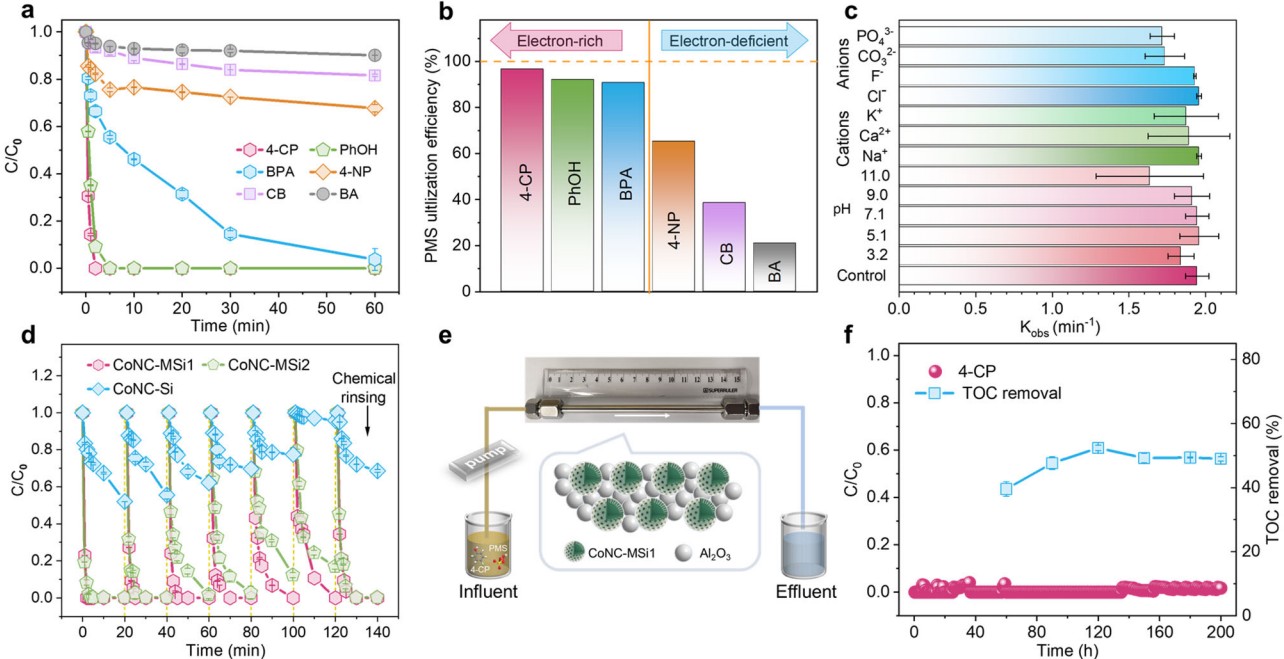

**Fig. 6 | Decontamination performances of the CoNC-MSi1 /PMS system under environmental-relevant conditions. a** Degradation performance and **b** PUE for removal of different organic pollutants including 4-CP, PhOH (phenol), BPA (bisphenol A), 4-NP (p-nitrophenol), CB (chlorobenzene), and BA (benzoic acid). **c** 4-CP degradation kinetics under different initial pHs and with environmentally relevant ions. **d** 4-CP removal efficiencies during cyclic operation. **e, f** Schematic diagram of a fixed-bed reactor loaded with CoNC-MSi1 catalyst and its performance for treating lake water spiked with 4-CP (initial TOC 43 mg L$^{-1}$, hydraulic retention time 3.24 min). Error bars of **a**–**d** and **f** represent the standard deviation, obtained by repeating the experiment twice. Reaction condition for **a**–**d**: [catalysts] = 0.25 g L$^{-1}$, [PMS] = 0.4 mM, [Pollutant] = 0.1 mM, [cations] = [anions] = 5 mM. Reaction conditions for **e, f**: [CoNC-MSi1] = 0.2 g, [Al$_2$O$_3$] = 1.3 g, [PMS] = 0.2 mM, [4-CP] = 0.05 mM.

correlation between the nanoconfinement effects and pore size in the SAC catalytic system remains elusive.

## Decontamination performances of the nanoconfined Co SACs in complicated water matrix

Attributed to the ETP-predominant pathway and the surface enrichment of PMS and pollutants, the CoNC-MSi1 exhibited extraordinary reactivity and PUE for Fenton-like oxidation of multiple electron-rich organic pollutants[64]. For example, besides 4-CP, a complete removal of phenol within 5 min, at a high PUE of 92.1%, was also achieved (Fig. 6a, b). In contrast, much lower degradation efficiency (5 - 22% removal within 60 min) and PUEs (21 - 65.3%) were shown by the electron-deficient compounds, including p-nitrophenol, chlorobenzene, and benzoic acid (Fig. 6a, b, Supplementary Fig. 31–32, and Table 8), confirming a high decontamination selectivity of the catalytic system. This feature rendered the CoNC-MSi1/PMS system superior environmental robustness[65], as evidenced by the negligible interferences by environmentally ubiquitous inorganic ions (Na$^+$, K$^+$, Ca$^{2+}$, Cl$^-$, F$^-$, CO$_3^{2-}$, and PO$_4^{3-}$), natural organic matters (NOM) and pH fluctuations (Fig. 6c and Supplementary Figs. 33–35). Here, the steric hindrance effect of the large-molecule NOM might also contributed to the superior NOM resistance of the nanoconfined SACs[20]. In addition, the system maintained high decontamination activity during four consecutive running cycles (Fig. 6d), although a slight performance decline occurred during prolonged operation due to catalyst passivation by the oxidation intermediates (Supplementary Fig. 36). The activity could be fully restored by eluting the catalyst with ethanol to remove the surface-accumulated intermediates, indicating a superior structural stability of the CoNC-MSi1. Its high stability was also supported by the unchanged material morphology and structure (Supplementary Fig. 37–39) as well as the negligible Co$^{2+}$ leaching (only 0.007 mg L$^{-1}$) after the decontamination reaction. Notably, although

the CoNC-MSi1/PMS system achieved only moderate 4-CP mineralization (~60% TOC removal within 60 min) (Supplementary Fig. 40), low-toxicity intermediates such as maleic acid, propanedioic acid, crotonyl alcohol, and acetic acid were formed, thus rendering a drastically reduced ecotoxicity of the effluent (Supplementary Fig. 41).

The potential of the CoNC-MSi1/PMS system for practical water purification was validated by the continuous-flow experiments in a packed-bed reactor fed with real lake water (Fig. 6e). The CoNC-MSi1 was loaded, together with Al$_2$O$_3$ fillers, into the reaction column and operated under pressurized condition. Remarkably, the catalytic system maintained superior decontamination activity and stability during 200-hour continuous operation, and can bear up to 10 MPa pressure attributed to the high-strength silica core of the catalyst. It steadily achieved complete 4-CP removal and 70% TOC removal at a short hydraulic retention time of only 3.24 min (Fig. 6f and Supplementary Fig. 42). A comparison with the TOC removal in 4-CP-containing pure water suggests that about half of the NOM in lake water was removed by the packed-bed reactor, likely mainly ascribed to rejection by the filling materials since NOM is resistant to surface oxidation in our system. Attributed to the superior pollutant selectivity, the catalytic system allowed much less PMS input (0.2 mM) than other SAC-based Fenton-like systems (typically 1–2 mM) for water purification[17,66]. The low biotoxicity of the treated lake water and negligible metal ions leaching (Supplementary Figs. 43 and 44) further confirm a high decontamination capability and environmental benignity of our nanoconfined catalytic system.

Given its superior selective decontamination performance, the CoNC-MSi1/PMS system may be readily integrated with other water treatment processes to facilitate practical application. Here, we show that the catalyst could be incorporated into an inorganic ceramic membrane to construct an efficient membrane catalytic decontamination. At water flux of 200 L m$^{-2}$ h$^{-1}$ and under an operating pressure

of 0.01 ± 0.003 MPa, this system maintained 100% 4-CP removal efficiency during 5-hour continuous operation (Supplementary Fig. 45), demonstrating a great potential for cost-effective water treatment application.

## Discussion

In this work, we unveil a fundamental catalytic pathway transition of Co SAC triggered by nanoconfinement- a phenomenon that has been ignored in previous nanoconfined SAC studies. In particular, a fine modulation of the catalytic pathway within the nonradical catalytic regime was achieved, and the underlying mechanisms were clarified. These findings significantly broaden our knowledge of nanoconfinement catalysis, which has been focused mainly on nanoscale catalysts previously[23]. Importantly, by leveraging the nanoconfinement effect to fine-tune the local electronic structure and chemical properties of the metal sites, highly efficient SACs could be constructed. The CoNC-MSi1/PMS system here exhibited unprecedentedly high activity and PUE for selective degradation of electron-rich pollutants, and demonstrated superior stability and environmental robustness to facilitate water treatment application. In addition, with a mediated ETP pathway, it may offer opportunities for the development of spatially decoupled pollutant oxidation and oxidant activation processes (i.e., requiring no direct contact between the oxidant and the water pollutants) to facilitate aquatic environmental remediation[67], thus avoiding the chemical wasting and ecological risks of conventional advanced oxidation technologies. Of course, the nanoconfinement strategy demonstrated here may also be readily extended to other SACs with further lowered costs and raised performance to favor practical application.

It has to be pointed out that selective decontamination technologies, like the nanoconfined catalytic system here, mainly target electron-rich pollutants and are less efficient for the electron-deficient ones (e.g., many halogenated organic pollutants). Thus, to boost its application in real environmental niches, an appropriate integration with conventional technologies such as adsorption, Fenton oxidation, and biological treatment would be necessary. For example, it may serve as a low-cost pretreatment to remove the majority of phenolic compounds in wastewater from refinery and chemical engineering industries, thereby lowering the working load and cost of downstream treatments. Or, it might be combined with an adsorption process to strengthen micropollutant removal from environmental water bodies or to afford deep purification of wastewater treatment effluent at minimal chemical consumption. All these possibilities warrant further investigation and demonstration.

## Methods

### Chemicals

All chemicals and reagents were of analytical grade and used without further purification. Potassium peroxomonosulfate (PMS, $H_3K_5O_{18}S_4$, 4.5% active oxygen), p-chlorophenol (4-CP), phenol (PhOH) and bisphenol A (BPA), p-nitrophenol (4-NP), Chlorobenzene (CB) and benzoic acid (BA), Hexadecyl trimethyl ammonium Bromide (CTAB), 2,3-dihydroxynaphthalene, 1,10-Phenanthroline, methylene blue (MB) and urea were purchased from Aladdin Bio-Chem Technology Co. Ltd. Ammonium hydroxide, Tetraethyl orthosilicate (TEOS), n-hexane, $Co(OAC)_2 \cdot 4H_2O$, Nitrotetrazolium Blue chloride (NBT), 1,3-Diphenylisobenzofuran (DPBF), Methyl phenyl sulfoxide (PMSO), Methyl phenyl sulfone(PMSO$_2$), Dimethyl sulfoxide (DMSO) and ethanol were purchased from Sinopharm Chemical Reagent Co., Ltd. $D_2O$, tert-butanol (TBA), furfural alcohol (FFA), Mobile phase (gradient-grade methanol and acetonitrile), acetone, sodium thiosulfate ($Na_2S_2O_3$) and spin trapping reagents (5,5-dimethyl-1-pyrroline-N-oxide (DMPO) and 2,2,6,6-tetramethyl-4-piperidone (TEMP)) were purchased from Sigma-Aldrich Co. Ltd. Carbon paper was purchased from Lige Science Co. (China) and tailored to 4.5 cm×4.5 cm in size.

### Synthesis of mesoporous silica particles

The mesoporous silica (MSi) particles were synthesized following the previously described protocols with slight modifications[68]. Specifically, 1 g $SiO_2$ particles with a mean diameter of ~300 nm were redispersed in 80 mL water solution containing 3 g CTAB and 0.9 g urea, and then 90 mL of n-hexane was added to form a biliquid phase system. To tune the pore size of the particles, the polarity of the upper organic solution was regulated by adjusting the n-hexane : toluene ratio to 1:0 for fabricating CoNC-MSi1 (average pore size of 7.7 nm), and to 1:1 for fabricating CoNC-MSi2 (average pore size of 12.8 nm). Subsequently, 2.76 mL of Isopropyl alcohol and 2 mL of TEO$_S$ were added dropwise into the solution slowly. All the synthesis procedures were carried out under gentle stirring (140 rpm) to maintain a biliquid phase solution. After a 12-h reaction at 70 °C, the product was collected and washed with ethanol and DI water. The solid samples were calcined in air for 3 h at a rate of 1 °C/min till 550 °C to remove CTAB templates. Then, the samples were added to 5% hydrochloric acid and stirred for 5 h, followed by thoroughly washing with ethanol and DI water. Lastly, the sample was vacuum-dried at 60 °C for 12 h before use.

### Synthesis of nanoconfined single-atom catalysts on MSi

The CoNC-MSi catalysts were synthesized by a facile two-step calcination method according to the literature with slight modifications[69] (Supplementary Fig. 1). Briefly, 2,3-dihydroxynaphthalene (0.18 g) was dissolved in 5 mL acetone; After adding MSi (0.1 g) and dispersion for 5-10 min, the mixture was stirred at 25 °C for 4 h. Then, the solid was collected by centrifugation, dried, and calcined in $N_2$ atmosphere at a series of temperatures to obtain MSi with graphitic carbon coating (C-MSi): 300 °C for 1 h, the temperature rose at a rate of 5 °C /min, and held at 800 °C for 2 h. Afterward, the C-MSi (0.1 g) was added into 100 mL ethanol solution containing $Co(OAC)_2 \cdot 4H_2O$ (0.36 g) and 1,10-Phenanthroline (1.26 g), dispersed for 5 min, and subjected to 4-hour reaction at 60 °C under reflux stirring. Then, the solid sample was dried and calcined in $N_2$ to 800 °C for 2 h at a rate of 10 °C/min. Lastly, the dried sample was added into 0.5 M $H_2SO_4$ and held for 6 h at 80 °C to ensure sufficient leaching of the Co clusters and unreacted Co ions, followed by thoroughly washing with DI water until neutral pH and drying to obtain CoNC-MSi For comparison, the unconfined CoNC-Si was also prepared by the same procedure as described above except for replacing the predecessor MSi with $SiO_2$ particles.

### Catalyst characterization and analysis

The morphologies of samples were examined by SEM (JEOL Co., U.S.A.) and TEM, using a Hitachi-7700 microscope with an accelerating voltage of 100 kV. Energy-dispersive X-ray spectroscopy (EDS) mapping was performed using a JEM-2100F field-emission high-resolution transmission electron microscope operated at 200 kV. Single-atom Co was characterized by AC-HAADF-STEM, and the images were captured using a JEOL JEM-2010 LaB6 high-resolution transmission electron microscope operated at 200 kV. The phase and composition of the materials were analyzed by XRD spectroscopy. The XRD patterns of samples were recorded on a Rigaku Miniflex-600 operated at 40 kV voltage and 15 mA current with Cu Kα radiation (λ = 0.15406 nm). The $N_2$ adsorption-desorption isotherms were obtained on a Micromeritics ASAP 2020 system to determine the Brunauer-Emmett-Teller (BET) specific surface area and pore size of catalysts. The XPS was collected on the scanning X-ray microprobe (PHI 5000 Verasa, ULAC-PHI, Inc.) using Al Ka radiation and the C 1$s$ peak at 284.8 eV as an internal standard. Raman spectra were measured using a Raman spectrometer (LabRAM HR Evolution, Horiba Co., Japan) equipped with 532 nm laser. FT-IR spectroscopy spectra were collected by IR microscopy (NicoletiN10, Thermo Fisher Inc.). The ESR signal spin-trapped by DMPO and TEMP were recorded on a Bruker spectrometer (A300, Bruker, Karlsruhe, Germany) with the following settings: center field=3512 G, microwave frequency=9.86 GHz, and power=6.36 mW.

The Co microstructire of the samples was investigated by employing the X-ray absorption spectrometry, including soft-XAS at BL12B-a beamline in the XAFS at BL14W station in SSRF (Shanghai, China).

### Evaluation of catalyst decontamination performance in batch experiments

The Fenton-like catalytic activity of CoNC-MSi1 for water treatment was evaluated using 4-CP as a model pollutant. The experiment was conducted in a 50 mL beaker, and the reaction solution (40 mL) contained: 0.1 mM pollutant (4-CP or phenol) and 0.25 g·L$^{-1}$ catalysts. The initial pH of the solution was adjusted to ~6.7 unless otherwise specified. After 30-min stirring to reach adsorption-desorption equilibrium, 0.4 mM PMS was added to initiate the reaction. The solution was magnetically stirred at 450 rpm under room temperature. Samples are collected at regular time intervals and immediately added with 200 mM $Na_2S_2O_3$ to cease the reaction prior to analysis. All the experiments were carried out in duplicate or triplicate. More details are provided in Supplementary material.

### Continuous-flow experiment for water treatment

A packed-bed reactor was constructed by using a home-made HPLC column (150 × 4.6 mm) filled up with 0.2 g CoNC-MSi1 microparticles and 1.3 g $Al_2O_3$ packing material. Before the test, the column was rinsed with DI water for 30 min. Then, the feed solution containing 0.05 mM 4-CP (DI water or lake water) and 0.2 mM PMS was continuously pumped into the column using a pre-pump (Lab Alliance), at a flow rate of 0.2 mL min$^{-1}$ and under pressure of ~10 MPa, to initiate the Fenton-like reaction. The samples were taken at given time intervals and immediately added with 200 mM $Na_2S_2O_3$ to cease the oxidation prior to analysis.

### Molecular dynamics simulations

The distributions of 4-CP and PMS molecules within the nanopores of the CoNC-MSi1 and CoNC-MSi2 and on the unconfined planer CoNC-Si were estimated by molecular dynamics simulations. Herein, the canonical ensemble model was selected and run for 5×106 steps with the time step of 1 fs. The temperature of simulated system was set as 298 K and controlled by a Nosé-Hoover thermostat with a coupling time constant of 0.1 ps[70]. The OPLS force field was chosen during the simulations and the cutoff was set as 14 A[71]. All simulations were performed by LAMMPS software[72].

### Spin-polarized DFT calculations

Spin-polarized DFT calculations were performed using the Materials Studio 7.0 CASTEP program. The generalized gradient approximation (GGA) with spin-polarized Perdewn-Burke-Ernzerhof (PBE) scheme was employed for calculating the exchange and correlation function[73,74]. The local density approximation (LDA + U) method was applied to correct the strong electronic correlation of Co 3$d$ electrons and the core electrons were represented by the OTFG ultrasoft pseudopotentials[75]. The kinetic energy cutoff and self-consistent field (SCF) tolerance were set at 571.4 eV and 1 × 10$^{-6}$ eV, respectively. The $k$-point in optimizing bulk models was 1 × 1 × 1. The structure optimization was performed with restriction of the catalyst edge for the stability of curved and planer structures. The $E_{ads}$ of PMS and 4-CP is defined as: $E_{ads} = E_{total} - E_{catalyst} - E_{PMS/4-CP}$, where the $E_{total}$, $E_{catalyst}$, and $E_{PMS/4-CP}$ represent the energy of catalyst/adsorbents system, the sole catalyst and PMS/ 4-CP molecule, respectively. Based on the EXAFS result, $CoN_4$-doped graphene model was used to stimulate PMS-SAC interactions on CoNC-MSi catalysts in this work. To simulate the catalysts with different pore sizes, the model of single-wall carbon nanotube with diameters of 7.7 nm and 12.8 nm was adopted, corresponding to the CoNC-MSi1 and CoNC-MSi2 catalyst, respectively. A planer $CoN_3$ and $CoN_4$ models were used

to simulate the CoNC-Si for evaluating the effect of the coordination environment on catalytic behavior. The free energies of the species were calculated as: $\Delta G = \Delta E + \Delta ZPE - T\Delta S$, where the $\Delta E$ was the change of reaction energy, the zero-point energy ($\Delta ZPE$) and entropy ($\Delta S$) were obtained from the calculation of vibration frequencies[76].

## Data availability

The authors declare that all the data supporting the findings of this study are available within the article and the Supplementary Information file. The source data of the figures are available on Figshare at https://figshare.com/s/f5550258d3cc93ef49e6.

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

## Acknowledgements

This work was supported by the National Natural Science Foundation of China (52192681 to W.W.L., U21A20160 to W.W.L., 51821006 to W.W.L., and 12275271 to C.W.), National Key Research and Development Program of Anhui Province (202104i07020003 to W.W.L.). We thank the BL14W1 in Shanghai Synchrotron Radiation Facility (SSRF) for help in XAFS characterizations, the Shiyanjia Lab (www.shiyanjia.com), Dr. Yaping Li from Center for Micro- and Nanoscale Research and Fabrication and Dr. Houqi Liu from Physical and Chemical Analysis Center at Suzhou Institute for Advanced Research, University of Science and Technology of China for help in other characterizations. The numerical calculations were performed on the supercomputing system in the Supercomputing Center of the University of Science and Technology of China.

## Author contributions

Y.M. came up with the original idea and carried out experiments and characterizations. Y.Q.L. performed the DFT calculations. C.W. performed the XAFS characterization and analysis. Y.S. and Y.J.W. gave suggestions and assistance on the manuscript. W.W.L. and Z.Y.G. directed the project and revised the manuscript. W.Q.X., T.L., X.C., and J.J.C. gave suggestions on experiments and discussed the data.

## Competing interests

The authors declare no competing interests.
