## [Peer Review File · Nature Communications]

Nanoconfinement Steers Nonradical Pathway Transition in Single Atom Fenton-like Catalysis for Improving Oxidant UtilizationREVIEWER COMMENTS

Reviewer #1 (Remarks to the Author):

In this work, the authors fabricated two Co-SACs based nanoconfined catalytic systems and one Co-SACs based control system for the investigation of the pollutant removal performance and mechanism. Interestingly, two performance and mechanism were observed. The catalytic mechanism changed from $1O_2$ to ETP pathway for the Co-SACs based control system ($1O_2$) and Co-SACs based nanoconfined catalytic systems (ETP pathway). Moreover, the catalytic performance was also significantly improved by the Co-SACs based nanoconfined catalytic systems. In-depth analyses using DFT, and MD were conducted to elucidate the mechanism. Overall, new insights could be obtained from this manuscript with interesting results. However, there are still a few issues (with some big issues) limiting the quality of this work and should be addressed.

1. Introduction. This statement of "nanoconfinement simply alters the kinetics of interfacial catalytic reactions" is not accurate. In fact, nanoconfinement also significantly changes the electronic interactions including electron density, free energy and electron transfer et al.
2. Introduction. grammar issue. rephrase this sentence. "Compared to the unconfined catalytic system, the nanoconfined SAC with fined-tuned pore size, the catalytic system achieved 34.7-fold faster"
3. Introduction. "Stable performance of the nanoconfined SAC system for almost complete removal of pollutants from lake water in a packed-bed reactor during the 200-hour continuous operation was also demonstrated ". The pollutants herein include NOM? Can the nanoconfined SAC system completely remove NOM? If not, this statement should be rephrased.
4. CoNC-MSN1 (0.07 wt%) than in CoNC-MSN2 (0.05 wt%) and CoNC-SN (0.01 wt%). It is confused about the Co SACs loading amount. At such low concentration?
5. What is the difference for CoNC-MSN1 and CoNC-MSN2. If the nanopore size can be adjusted, why not fabricating catalyst with smaller pore size as the current pore sizes of 7.7 nm and 12.8 nm are large in terms of nanoconfinement effect-the smaller spaces (especially for subnanometer spaces), the more obvious effect.
6. "The nanopore structure offers the nanoconfined catalysts 1010-times larger specific surface area than the nonporous CoNC-SN for anchoring of Co single atoms." How did the authors make this conclusion? Based on the BET surface area, there is only 3 times higher for the nanoconfined catalysts compared to the control.
7. Fig. 1c, show the pore size distribution in SI rather than just giving the mean pore size.
8. Big issue. "Normalizing the degradation kinetic constant to specific surface area and the catalysts dosage still yields 10.3-fold higher specific activity for CoNC-MSN1 ($k_{SA} = 0.154 \text{ L m}^{-2} \text{ min}^{-1}$) than CoNC-SN (Fig. 3C)." Considering that the effect catalyst is Co SAC rather than the hybrid catalyst CoNC-MSN and also the Co SAC loading amount for three catalysts is different, whether the normalized K is reasonable in the way that $K/\text{surface area} \cdot \text{dosage}$?
9. Fig. 2c, the Co foil and CoO references should be included in the figure for comparison.
10. Big issue. Based on Co K-edge XANES and EXAFS spectra, the Co valences between 0 and 2 for all the three catalysts. However, there are only Co^{3+} , Co^{2+} species for CoNC-MSN catalysts with only about 3% Co^0 for the control based on XPS results-Supplementary Figure 8. How to explain this conflict?
11. Big issue. Fig. 3. What is the mineralization rate for these systems? In terms of water decontamination, the intermediates of 4-CP are also important. If the mineralization rate is low with the generation of lots of intermediates, then the toxicity assessment is very important to check whether the proposed AOP indeed achieved decontamination. However, these results are currently missing in this work.
12. Fig. 3D, what is the effect of MB herein? for quenching which ROS?
13. Fig. 4E, what is the experimental period and how about the current change during the process?
14. Big issue. Fig. 5I. It seems like PMS and 4-CP distribution concentrations are highest at about 0.5-1 nm. However, how does this relate to the pore size with the mean values of 7.7 nm and 12.8 nm?
15. Big issue. The authors are strongly recommended to check their submitted manuscript. All figures

should be self-explained. Meanwhile, Fig. 5 was partially displayed. What is the meaning of the X and Y axis titles of Fig. 5I?

16. In addition, it retained high activity within the pH range of 3-11. Is the PH stable during the AOP process? At least the PH change during the whole reaction period should be given for one group of the results shown in Fig. 2.

17. How to understand the increased TOC removal with time shown in Supplementary Fig. 35?

18. How about the chemical state of Co in the used catalyst, which is important to show its stability. As such, XPS analysis should be supplemented for the catalyst after usage, however, which is missing.

19. Big issue. Fig.6. In terms of application, there will be numerous pollutants including the electron-rich and the electron-deficient. Herein, the ETP mechanism is efficient in the removal of electron-rich pollutants rather than the electron-deficient with a low mineralization rate. However, water decontamination does not only mean pollutant removal and also mineralization. More importantly, toxicity reduction including feed and the generated intermediates should be ensured in terms of water decontamination, which however is missing herein. As such, the shortage of this technique should also be mentioned and the evaluation of toxicity should be supplemented.

20. Fig. 6E, how about the particle size of Al₂O₃? Will these Al₂O₃ fillers facilitate PMS activation? For the packed-bed reactor, how to make sure there will be no leaching of catalyst and Al₂O₃ fillers?

Reviewer #2 (Remarks to the Author):

Review Comments

Title: Nanoconfinement Steers Nonradical Pathway Transition in Single Atom Fenton-like Catalysis for Improving Oxidant Utilization

My opinion- Reject

General Comments:

The work described here doesn't meet the publication standards typically associated with journals like Nature Communications. The three major reasons are i) the material prepared in this study doesn't have any novel aspect. ii) The results on nanoconfinement is not satisfactory based on the results provided, and iii) The chosen problem and its outcomes lack interest and do not offer novel insights in this field. The quality of discussion throughout the manuscript needs an improvement. In some place's authors used the scientific terms that are not defined. Therefore, this manuscript does not meet the required standard. A very few comments that are provided below.

1. After reading this manuscript, it invites several fundamental questions. i) Why does authors choose such a complicated synthetic procedure that involves organic solvents from toluene to n-hexane having multiple reactions steps. ii) Even after a hard synthesis procedure, it is disappointing to see a surface area of 50 m² g⁻¹ maximum for a catalyst. iii) After all, I didn't see any positive aspect in using this catalyst for degradation because it doesn't give any advantage compared to already reported PDS/PMS activation catalysts. The degradation performance showed in this paper are almost identical to some of the carbon materials like activated carbon reported already.

2. By looking at the particle size of MSN (300 nm), CoNC-MSN1 (400 nm), and CoNC-MSN2(400 nm) are they really nanoparticles? Throughout the manuscript this terminology radiates.

3. Under what conditions CoNC-MSN1 (400 nm), and CoNC-MSN2 (400 nm) are achieved. It is not mentioned in the material synthesis section.

4. First of all, in page.3, TEM analysis is not suitable for accurately measuring the pore size of any material. This sentence needs to be revised.

5. The authors assert an average nanopore size within the range of 7.7 to 12.8 nm. To our surprise, no mesoporous features were noticed in the BET data. As this pore size falls within the mesoporous scale type-IV isotherm is missing why?
 6. In page. 3 what does the 1010-times larger surface area mean. There is no such an increase in BET results (Table S1).
 7. In page.3 The statement "stabilization of Co single atom by the Si-OH groups" is contradictory with "a typical feature of Co SAC with N coordination" in the next paragraph. The claims made should be consistent throughout the manuscript.
 8. If N-atoms from carbon are the coordination sites of Co single-atoms. What is the role of SiO₂ here. Why it is not removed through etching?
 9. It is apparent that results of O₂ suppression from CoNC-SN to CoNC-MSN₂, and CoNC-MSN₁ is well correlated with Co concentration in the system, rather than the nanoconfinement effect claimed by authors. The kobs in Fig. 3B agrees well with it.
 10. The quenching effect observed with FFA is debatable in PMS activation system (<https://doi.org/10.1016/j.chemosphere.2023.138264>). Also, the inhibition can be caused by FFA oxidation itself sometimes.
 11. Where is DPBF in Fig. 3D?
 12. Fig. 5 appears incomplete as only half of it is visible. Authors are advised to exercise caution during the uploading and approval of submissions to ensure the figures are fully visible.
 13. Many important publications in this field are missing. For eg. 10.1038/s41467-022-30560-9,
 14. In page. 7 what is mean by "as evidenced by its high NaClO₄ resistance³"?
 15. Is this the electrical conductivity measuring device? The authors didn't used the work electrochemical impedance spectroscopy.
- There are many such issues in this manuscript that can be pointed out in every paragraph. I recommend authors should draft the manuscript carefully before submitting to any journal.

Specific comments:

In Page.3 There is a typo in (Fig. 1E-F)

In Page.4 There is a typo in "tothe"

The rate constant symbol kobs, is used in different ways Kobs (Fig. 3), kobs (page.5)

Define PUE in the manuscript.

In page.7 there is a typo 'peak at ~835 cm⁻¹'

Reviewer #3 (Remarks to the Author):

This work reports an interesting nanoconfinement effect that has been overlooked in previous studies. Nanoconfined reactions are actually widespread in pollution control systems that applying porous nanomaterials and deserve attention. Although there are already several studies about application nanoconfined single atom catalysts in Fenton-like oxidation, this might be the first report of nanoconfinement-triggered catalytic pathway transition occurring in single atom catalysis system. Especially, while pathway change from radical to nonradical reactions have been reported in Fenton-like systems, a transition between different nonradical pathways is scarce. It also provides new insights to benefit a better understanding of the boosted reaction kinetics in nanoconfined catalysis, which has been widely believed to be primarily ascribed to accelerated interfacial mass transfer. The good performances of the catalytic system with fine-tuned nanoconfinement degree for selective decontamination and stable wastewater treatment operation were demonstrated. The major conclusions are mostly well supported by experimental evidences and theoretical calculations. Overall, I think the current work is of scientific value and practical significance to boost the development of Fenton-like oxidation technologies toward practical application. However, one figure is incompletely displayed and it would be helpful for the readers to better understand this work by clarifying the following issues with a minor revision.

1. What are the specific advantages of the catalytic pathway transition from singlet oxygen to the electron-transfer process? How is it related to the saving in oxidant consumption? The benefits of such pathway transition in practical decontamination applications should be clarified in more details.
2. In the 4-CP degradation reaction, the performances of catalysts were compared based on different dosages (mass concentration or Co content basis in different cases). Why not using unified dosage? This inconsistency should be addressed or explained.
3. To substantiate the advantages of the nanoconfined catalytic system, it would be helpful to supplement experimental evidences about the influence of inorganic anions or organic compounds in the other two control groups.
4. The figure quality should be improved. For example, the display of Fig. 5 is not complete; what's meaning of the solid line and the dashed line in Figure 4G? The full names of the pollutants should be provided in the figure caption; the font format across all figures should be standardized in the manuscript. A comprehensive refinement of the graphical representations is advised to ensure the consistency and clarity.
5. The English language can be further improved for better clarity and readability. The format of references should be unified following the guidelines.

Reponses to Reviewer #1's Comments

In this work, the authors fabricated two Co-SACs based nanoconfined catalytic systems and one Co-SACs based control system for the investigation of the pollutant removal performance and mechanism. Interestingly, two performance and mechanism were observed. The catalytic mechanism changed from $^1\text{O}_2$ to ETP pathway for the Co-SACs based control system ($^1\text{O}_2$) and Co-SACs based nanoconfined catalytic systems (ETP pathway). Moreover, the catalytic performance was also significantly improved by the Co-SACs based nanoconfined catalytic systems. In-depth analyses using DFT, and MD were conducted to elucidate the mechanism. Overall, new insights could be obtained from this manuscript with interesting results. However, there are still a few issues (with some big issues) limiting the quality of this work and should be addressed.

Response: We appreciate the reviewer's positive comments and constructive suggestions, which are very helpful for guiding the revision of our manuscript. All the major concerns of the reviewer have been addressed (detailed below).

1. Introduction. This statement of "nanoconfinement simply alters the kinetics of interfacial catalytic reactions" is not accurate. In fact, nanoconfinement also significantly changes the electronic interactions including electron density, free energy and electron transfer et al.

Response: Thank you for pointing out the inaccurate expression. The sentence has been rephrased following your comment.

Revision:

".....Recently, attempts have also been made to confine the SAC active sites within a nano-structured space, which was found to drastically raise the local concentration of reactive species and pollutants^{1, 2}, enabling 2~3 order-of-magnitude higher activities for Fenton-like oxidation of pollutants than the unconfined analog^{3, 4}. Nanoconfinement could also alter the physicochemical properties of metal catalytic sites, thereby affecting the thermodynamics of surface-reactant interactions and enhancing or

altering the product selectivity and even triggering catalytic pathway shift^{5, 6}.” (Page. 2)

2. Introduction. grammar issue. rephrase this sentence. "Compared to the unconfined catalytic system, the nanoconfined SAC with fine-tuned pore size, the catalytic system achieved 34.7-fold faster"

Response: The sentence has been rephrased and the Introduction section has been thoroughly rewritten to improve the language.

Revision:

“.....Our results not only validate the nanoconfinement-induced reactant enrichment effect but also unveil a changed pollutant degradation pathway from 1O_2 to electron transfer process (ETP) triggered by nanoconfinement. Such changes together render the catalytic system significantly strengthened activity in 4-CP degradation (up to 34.7-fold higher rate kinetic constant than the unconfined control) and raised PMS utilization efficiency (PUE from 61.8% to 96.6%).” (Page. 3)

3. Introduction. "Stable performance of the nanoconfined SAC system for almost complete removal of pollutants from lake water in a packed-bed reactor during the 200-hour continuous operation was also demonstrated ". The pollutants herein include NOM? Can the nanoconfined SAC system completely remove NOM? If not, this statement should be rephrased.

Response: We apologize for the unclear expression. **NOM was present in the lake water but was not considered as a pollutant here.** In this work, we mainly target the removal of toxic, recalcitrant organic compounds, although a partial removal of NOM from lake water was also achieved. A comparison of TOC removal in 4-CP-containing pure water and lake water (with NOM) showed that **about half of the NOM in lake water was removed by the packed-bed reactor** (Fig. 6F), in contrast to the almost complete removal with ~70% mineralization of 4-CP (Supplementary Fig. 41). Here, low removal of NOM should be mainly attributed to **rejection by the filling materials,**

and the selective oxidation nature of the catalytic system and the steric hindrance of the large-molecule NOM make it difficult for oxidative degradation of NOM. The relevant statement has been rephrased as suggested.

Revision:

Figure 6F. The performance for treating lake water spiked with 4-CP in a fixed-bed reactor system. Reaction condition: [catalysts] = 1.5 g, [PMS] = 0.2 mM, [4-CP] = 0.05 mM.

“.....*The salinity resistance, environmental robustness, and long-time stability of CoNC-MSi catalyst and its successful application for treatment of contaminated lake water in a continuous-flow packed-bed reactor were also demonstrated.*” (Page. 3)

4. CoNC-MSN1 (0.07 wt%) than in CoNC-MSN2 (0.05 wt%) and CoNC-SN (0.01 wt%). It is confused about the Co SACs loading amount. At such low concentration?

Response: We apologize for the confusing information about the Co SACs loading. For estimation of SAC loading, the content of single atoms is typically normalized to the mass weight of the entire SAC material including the support, therefore the mass density of the support material matters. In previous studies, the Co-SAC commonly adopted carbon support with low mass density, rendering them Co loading in the range of 1~5 wt% or 0.2~1 at.% Co (*Nat. Chem.* 2021, 13, 887–894). However, in our work the core-shell structured porous silica particle with high mass density was used as the

support, thus resulting in a seemingly much lower SAC loading. Actually, if we only take into account the shell layer that harbors the Co atoms (measured by peeling off the shell from the SiO₂ core via ultrasound treatment), we get a calibrated Co loading of 3.8 wt% for CoNC-MSi1 (Supplementary Table 2). Therefore, **the actual Co loading of the catalyst in our study was comparable with those in previously-reported SACs.** The relevant explanation has been supplemented in the revised manuscript.

Revision:

Supplementary Table 2. The Co contents relative to the shell or to the total material of different catalysts.

Catalyst		CoNC-MSi1	CoNC-MSi2	CoNC-Si
Co content ^a	in the shell	3.8 ± 0.3	1.2	0.2 ± 0.1
(wt%)	in the total material	0.07	0.05 ± 0.01	0.01

a. All the Co loading amount was assessed twice by ICP-AES measurement.

Note: The core-shell structured porous silica particle with high mass density was used as the support and Co single atoms were confined on the porous shell, thus resulting in a seemingly much lower SAC loading.

5. What is the difference for CoNC-MSN1 and CoNC-MSN2. If the nanopore size can be adjusted, why not fabricating catalyst with smaller pore size as the current pore sizes of 7.7 nm and 12.8 nm are large in terms of nanoconfinement effect-the smaller spaces (especially for subnanometer spaces), the more obvious effect.

Response: We are grateful for the reviewer's constructive suggestion. The CoNC-MSi1 and CoNC-MSi2 (renamed here for more accurate expression) differed mainly in the nanopore size (7.7 nm v.s. 12.8 nm), and accordingly, the surface area available for Co loading varied. It is true that the pore size critically affects the nanoconfinement effects and smaller sizes are supposed to yield more obvious nanoconfinement effects in theory. However, **the actual scenario could be more complicated, because the nanoconfinement could fundamentally alter the chemical and physical properties of the confined reaction molecules and the catalytic sites, which are all size-**

dependent. This leads to uncertain impacts on the nanoconfined reactions and processes. For example, a previous study suggested that the Co-TiO_x laminate membrane with ultra-small channels exhibited much higher catalytic activity than the larger-sized ones for water contaminant degradation (*Nat Commun.* 2022 13, 4010). However, other studies showed much higher ion diffusion rate for the membranes with larger-sized nanochannels (*Adv. Mater.* 2020, 32, 2003251). In our work, the DFT calculations also suggest that, when further downsizing the nanopores from 7.7 nm to 2.7 nm, the catalyst shows sharply decreased adsorption energy (E_{ads}) for PMS and 4-CP binding, implying a declined catalytic activity under highly-nanoconfined condition (Supplementary Figure 29, and Table 7). Therefore, the size-dependent nanoconfinement effects in heterogeneous Fenton-like catalysis are not straightforward and remain poorly understood so far.

Here, we aim to clarify whether and how nanoconfinement would cause catalytic pathway change in heterogeneous decontamination systems by comparing the nanoconfined and unconfined groups. As for the complicated size-dependent nanoconfinement effects, we are planning to explore further into this issue in future studies.

Revision:

Supplementary Figure 29. Models and adsorption energy of PMS and 4-CP binding on nanoconfined catalysts with different pore size (7.7 nm v.s. 2.7 nm).

Note: The adsorption energy (E_{ads}) of PMS and 4-CP both decreased sharply when the pore size of catalysts decreased from 7.7 nm to 2.7 nm, implying that the proper nanoconfinement degree of CoNC-MSi1 for its optimal catalytic activity.

“.....However, according to the DFT calculation, further downsizing to 2.7 nm nanopores would weaken the PMS and pollutant binding instead (Supplementary Fig. 29 and Table 7), suggesting that the size-dependent nanoconfinement effects are complicated and warrant further clarification^{3,7}.” (Page. 10)

6. "The nanopore structure offers the nanoconfined catalysts 10^{10} -times larger specific surface area than the nonporous CoNC-SN for anchoring of Co single atoms." How did the authors make this conclusion? Based on the BET surface area, there is only 3 times higher for the nanoconfined catalysts compared to the control.

Response: Thank you for pointing out this issue. We agree that the conclusion of “ 10^{10} -times larger specific surface area”, drawn from a theoretical estimation of the specific surface area of an ideal MSi sphere, is inaccurate. Accepting the reviewer’s suggestion, we have re-measured the N_2 adsorption-desorption curves, which revealed 6.8-fold larger specific surface area of the CoNC-MSi1 than the nonporous counterpart (Fig. 1C, Supplementary Table 1). Therefore, we have corrected the statement based on the actual measurement data.

Revision:

Figure. 1C. N₂ adsorption-desorption isotherms and the pore size distribution curves in the illustration of two CoNC-MSi and CoNC-Si catalysts.

“.....The N₂ adsorption-desorption isotherms and the pore size distribution curves reveal different pore sizes of the two nanoconfined catalysts (Fig. 1C, Supplementary Table 1). The catalysts with average pore sizes of 7.7 nm and 12.8 nm were referred to as CoNC-MSi1 and CoNC-MSi2, respectively (Supplementary Table 1). The relatively small pore sizes of the CoNC-MSi1 were in line with its more densely-distributed nanopores in the shell, as manifested by the TEM mages (Fig. 1B, Supplementary Fig. 2). The porous structure also endowed the CoNC-MSi1 with 6.8-fold larger specific surface area than that of the nonporous CoNC-Si (Supplementary Table 1).” (Page. 3)

7. Fig. 1c, show the pore size distribution in SI rather than just giving the mean pore size.

Response: Accepting your suggestion, we have supplemented the N₂ adsorption-desorption isotherms and pore size distribution curves in **Fig. 1C**.

Revision:

Figure. 1C. N₂ adsorption-desorption isotherms and the pore size distribution curves in the illustration of two CoNC-MSi and CoNC-Si catalysts.

Supplementary Table 1. BET surface area, pore diameter, and pore volume of the

prepared samples.

Catalyst	BET surface area ($\text{m}^2 \text{g}^{-1}$)	Pore diameter (nm)	Pore volume ($\text{cm}^3 \text{g}^{-1}$)
CoNC-MSi1	85.89	7.71	0.15
CoNC-MSi2	74.27	12.83	0.21
CoNC-Si	14.48	/	/

“..... The N_2 adsorption-desorption isotherms and the pore size distribution curves reveal different pore sizes of the two nanoconfined catalysts (Fig. 1C, Supplementary Table 1). The catalysts with average pore sizes of 7.7 nm and 12.8 nm were referred to as CoNC-MSi1 and CoNC-MSi2, respectively (Supplementary Table 1).” (Page. 3)

8. Big issue. “Normalizing the degradation kinetic constant to specific surface area and the catalysts dosage still yields 10.3-fold higher specific activity for CoNC-MSN1 ($k_{SA} = 0.154 \text{ L m}^{-2} \text{ min}^{-1}$) than CoNC-SN (Fig. 3C).” Considering that the effect catalyst is Co SAC rather than the hybrid catalyst CoNC-MSN and also the Co SAC loading amount for three catalysts is different, whether the normalized K is reasonable in the way that $K/\text{surface area} \cdot \text{dosage}$?

Response: Thank you for pointing out this issue. We agree that the definition of the specific activity as $k/\text{surface area} \cdot \text{dosage}$ is inappropriate, and has corrected it to $k_{\text{obs}}/\text{specific surface area}$ (k_{SA}), the most widely accepted definition in the literature. In addition, to better reveal the intrinsic catalytic activity of the Co single atoms, we also provided the information on **activity per mass of Co sites ($k_{\text{per-site}}$) by normalizing the degradation rate constants to the Co loading amount** (Supplementary Fig. 11). The results show the highest $k_{\text{per-site}}$ value for CoNC-MSi1 and the lowest for CoNC-Si, consistent with the k_{SA} results (Fig. 3C) and the result of turnover frequency (TOF) (Supplementary Fig. 12). These results confirm that improved activity of the CoNC-MSi1 was ascribed to not only increased Co loading but also a raised intrinsic activity of the Co single atoms under nanoconfinement.

Revision:

Supplementary Figure 11. The 4-CP degradation rate constants normalized by total Co mass ($k_{\text{per-site}}$) of different systems.

Figure 3C. Comparison with other catalysts in PUE and specific activity for oxidation of pollutants.

“.....Further normalizing the degradation kinetic constant to specific surface area (k_{SA}) yields 10.3-fold higher specific activity of CoNC-MSi1 than CoNC-Si (Fig. 3C), and normalizing to Co loading amount ($k_{\text{per-site}}$)⁸ still shows 5.1-fold higher intrinsic reactivity of single Co atoms (Fig. 3C, Supplementary Fig. 11). Consistent results were obtained for the estimated turnover frequency (TOF; 4-CP degradation rate at per Co atom basis)⁹ (Supplementary Fig. 12). All these evidences strongly support a profoundly

raised intrinsic reactivity of the Co atoms under nanoconfinement, together with the increase Co loading, contributed to the accelerated 4-CP degradation.” (Page. 6)

9. Fig. 2c, the Co foil and CoO references should be included in the figure for comparison.

Response: Accepting the reviewer’s suggestion, we have added the Co K-edge FT-EXAFS spectra of the Co foil and CoO references in Fig. 2C. The result confirms the presence of Co-N first-shell coordination but an absence of Co-Co and Co-O coordination structure in both the CoNC-MSi1 and CoNC-MSi2.

Revision:

Figure.2C. Co K-edge FT-EXAFS spectra and of the Co SACs and the references (Co foil, CoO, and CoPC).

“.....In addition, a distinct peak of the Fourier-transformed (FT) k^3 -weighted EXAFS spectra at 1.4-1.6 Å (corresponding to the Co-N first-shell coordination) (Fig. 2C) and a wavelet transform (WT) peak at 5 Å⁻¹ (corresponding to the Co-N bond) (Fig. 2D–H) were detected for all the catalysts^{10, 11}. According to the EXAFS fitting results (Supplementary Table 5 and Fig. 7), the Co-N₄ dominated in the nanoconfined Co SAC, against the Co-N₃ coordination of the unconfined control, indicating a changed coordination environment of the Co single atoms under nanoconfinement¹². Notably, Co-Co signal was also detected solely in the CoNC-Si by the EXAFS and Co 2p XPS spectra (Fig. 2C, Supplementary Fig. 8), indicating a slight agglomeration of the

unconfined Co atoms although it counted for only ~3 wt % of the total surface Co contents¹³. Overall, the above results confirm a N-coordinated Co atom configuration in the catalysts and reveal an altered N coordination number under nanoconfinement (Fig. 2I).” (Page. 4)

10. Big issue. Based on Co K-edge XANES and EXAFS spectra, the Co valences between 0 and 2 for all the three catalysts. However, there are only Co³⁺, Co²⁺ species for CoNC-MSN catalysts with only about 3% Co⁰ for the control based on XPS results- Supplementary Figure 8. How to explain this conflict?

Response: Thank you for pointing out this issue. Indeed, the XPS and XAFS results were inconsistent in our previous manuscript. Therefore, we have re-analyzed the XPS spectra of the three catalysts (Supplementary Fig. 8a). The XPS spectra fitting results clearly show peaks of Co 2p_{3/2} (781 eV) and Co 2p_{1/2} (796 eV) corresponding to Co²⁺. In addition, a weak XPS signal of Co⁰ was also detected in the CoNC-Si. Now, the XPS results of Co valence are consistent with the EXAFS analysis. We have updated the XPS results and the relevant discussions in the manuscript.

Revision:

Supplementary Figure 8a. XPS spectra of Co 2p spectrum for different catalysts.

Note: The Co 2p XPS spectra of three catalysts all exhibit Co 2p_{3/2} peak¹⁴. Specifically, the binding energies of ~781 and 796 eV could be assigned to Co 2p_{3/2} and Co 2p_{1/2}

peaks, respectively, suggesting the presence of Co^{2+} in the three catalysts. The Co^0 peak (778.8 eV) only appeared in CoNC-Si, with Co^0 content accounts for only ~3% of the total amount. The spectra of C 1s and O 1s show distinct signals of C-C, C-N and C=O, indicating a successful loading of the carbon layers^{15, 16}.

11. Big issue. Fig. 3. What is the mineralization rate for these systems? In terms of water decontamination, the intermediates of 4-CP are also important. If the mineralization rate is low with the generation of lots of intermediates, then the toxicity assessment is very important to check whether the proposed AOP indeed achieved decontamination. However, these results are currently missing in this work.

Response: Thanks for your valuable suggestion. We have added the pollutant mineralization data (i.e., TOC removal) to more comprehensively show its decontamination performance (Supplementary Fig. 39). The two nanoconfined catalytic systems achieved 100% 4-CP degradation and 50~ 60% TOC removal within 60 min, indicating a moderate mineralization efficiency of the catalytic system. The LC-MS results further confirm the decomposition of 4-CP through open-ring reactions, forming small molecule products including maleic acid, propanedioic acid, crotonyl alcohol, and acetic acid (Supplementary Fig. 35). In contrast, the unconfined CoNC-Si achieved only ~22% mineralization of 4-CP within 60 min.

In addition, we added a toxicity test following your suggestion. Consistent with the pollutant degradation results, the nanoconfined catalytic systems achieved more profound toxicity reduction than the unconfined control. The toxicity of the water samples was assessed by measuring the cell viability of *E. coli* bacterium after 6-h exposure (*Sci Rep.* 2021, 11, 15978; *Nat Rev Microbiol.* 2011, 9, 779–790). The feed water groups showed 100% bacterial lethality, confirming a high acute toxicity of 4-CP. In comparison, the *E. coli* cells still remained 96% viability after exposure to the treated water from the CoNC-MSi/PMS systems (Supplementary Fig. 40), confirming that the catalytic treatment drastically reduced the water toxicity. In contrast, the CoNC-MSi/PMS treatment group showed only 43% bacterial viability, consistent with its

relatively low decontamination efficiency.

To address the reviewer's concerns, we have added the above information and relevant discussion in the manuscript.

Revision:

Supplementary Figure 39. TOC removal in different catalytic systems. Reaction conditions:

[catalysts] = 0.25 g L⁻¹, [PMS] = 0.4 mM, [4-CP] = 0.1 mM.

Supplementary Figure 40. The toxicity assessment of different reaction systems. (a) *E. coli* colony after 6-h cultivation in different water samples, (b) relative cell viabilities, and (c) growth curves in different systems.

Note: The toxicity of the water samples was assessed by measuring the cell viability of *E. coli* bacterium after 6-h exposure^{17, 18}. The results show 100% lethality of the cells after exposure to the initial reaction solution, confirming a high acute toxicity of 4-CP. After treatment, *E. coli* cells still remained 96% viability for the CoNC-MSi/PMS systems, indicating >90% toxicity reduction after the treatment. In contrast, only 43% toxicity reduction of the treated water was achieved by the CoNC-Si/PMS system, due to its relatively low decontamination efficiency. Therefore, the two CoNC-MSi/PMS systems show a detoxification effect and can be promising to transfer the biotoxic organic 4-CP to less toxic or harmless products.

“..... Notably, although the CoNC-MSi1/PMS system achieved only moderate 4-CP mineralization (~60% TOC removal within 60 min) (Supplementary Fig. 39), low-toxic intermediates such as maleic acid, propanedioic acid, crotonyl alcohol and acetic acid were primarily formed, thus the water ecotoxicity can be drastically reduced (Supplementary Fig. 40). These results highlight a great potential of the CoNC-MSi1/PMS system for practical water treatment application.” (Page. 12)

12. Fig. 3D, what is the effect of MB herein? for quenching which ROS?

Response: Methylene blue (MB) was used as a chemical probe to detect free radicals including hydroxyl ($\cdot\text{OH}$), sulfate radicals ($\text{SO}_4\cdot^-$) and surface-bound radicals (*Nature Communications*. 2021, 12, 4777; *Environ. Sci. Technol.* 2022, 56, 1, 564–574). The relevant information has been added in the main text and the figure caption (Fig. 3F).

Revision:

Figure 3F. Effects of different scavengers on 4-CP degradation kinetics and the fraction of $^1\text{O}_2$ pathway for decontamination. EtOH, TBA, MB, and FFA were used to detect $\cdot\text{OH}$, $\text{SO}_4^{\cdot-}$, surface-bound radicals, and $^1\text{O}_2$, respectively.

13. Fig. 4E, what is the experimental period and how about the current change during the process?

Response: We appreciate the reviewer's valuable suggestions, which motivate us to dig deep into the ETP reactions. The reaction period of the electrochemical assay was 3 hours (Supplementary Fig. 21a). To address the reviewer's concern, we have repeated the dual-chamber galvanic cell experiment, with the catalyst loaded on both electrodes, and monitored the current changes during the reaction (Supplementary Fig. 21b). The results show a sharp rise and rapid decline of current after adding 4-CP into the CoNC-MSi1 system, indicating strong adsorption and electronic interaction between 4-CP and the catalysts. Subsequently, the PMS addition caused a sharp rise of current again and held relatively stable thereafter, indicating a continuous reaction between the surface-bounded PMS and 4-CP mediated by the catalyst. In contrast, the CoNC-MSi2 group showed similar but much weaker current changes upon the subsequent addition of 4-CP and PMS, while the CoNC-Si groups exhibited the lowest responses, consistent with their different catalytic and decontamination activities. The relevant descriptions have been supplemented in the revised manuscript.

Revision:

“..... To elucidate the specific catalyst-pollutant-PMS interactions in the nanoconfined catalytic system, we constructed a galvanic cell, with the two electrolyte chambers (containing 4-CP and PMS respectively) being separated by a catalyst-loaded carbon flake. This system showed distinct current changes upon 4-CP and PMS addition accompanied by their synchronous decomposition in individual chambers (Fig. 4D–E, Supplementary Fig. 21), suggesting the existence of mediated ETP pathway for pollutant degradation. This was also supported by the obviously raised open-circuit potential with reactant concentration¹⁹ (Fig. 4F, Supplementary Fig. 22).” (Page 8)

Supplementary Figure 21. (a) 4-CP degradation performance without and with PMS and (b) current changes of the dual-chamber galvanic cells with different catalysts loaded on the electrodes Reaction condition: [catalyst] = $0.5 \text{ g} \cdot \text{L}^{-1}$, [PMS] = 0.8 mM, [4-CP] = 0.05 mM.

Note: To record current change of the dual-chamber galvanic cell in response to 4-CP and PMS addition, the Agilent 34970A data acquisition switch was employed. The results show a sharp rise and rapid decline of current after adding 4-CP into the CoNC-MSi1 system, indicating a strong adsorption and electronic interaction between 4-CP and the catalysts. After further adding PMS, the current showed sharp increase again and held relatively stable thereafter, indicating a continuous reaction between the surface-bounded PMS and 4-CP mediated by the catalyst. In contrast, the CoNC-MSi1 group showed similar but much weaker current changes upon subsequent addition of

4-CP and PMS, while the CoNC-Si groups exhibited lowest responses, consistent with their different catalytic and decontamination activities.

14. Big issue. Fig. 5I. It seems like PMS and 4-CP distribution concentrations are highest at about 0.5-1 nm. However, how does this relate to the pore size with the mean values of 7.7 nm and 12.8 nm?

Response: Thanks for your valuable suggestion. The MD calculation results show that, for all three catalysts, the distribution concentrations of PMS and 4-CP are both the highest at about 0.5-1 nm from the catalyst surface. However, nanoconfinement does raise the local concentrations of the reactants (Fig. 3D–E). Compared with the CoNC-Si, the CoNC-MSi1 exhibited 1.8-fold and 3.9-fold higher peak concentrations of PMS and 4-CP respectively, indicating **an obvious enrichment of reactants induced by nanoconfinement**. In comparison, the CoNC-MSi2 showed only 2.1~2.4-fold higher peak concentrations of reactants than the CoNC-Si, indicating a size-dependent enrichment effect for the nanoconfined catalysts. Therefore, higher-degree nanoconfinement can raise the intrinsic activity of reactive sites and strengthen reactant enrichment, thereby profoundly affecting both the kinetics and even thermodynamics of the surface reactions (*Adv. Mater.* 2020, 32, 2001068; *ACS EST Engg.* 2021, 1, 4, 706–724). The relevant descriptions have been supplemented in the revised manuscript.

Revision:

Figure 3 D–E. Radial distribution function of PMS and 4-CP distribution concentration

of different catalysts obtained by molecular dynamic simulation.

“..... The raised specific activity and PUE of nanoconfined Co SAC for pollutant degradation could be partially ascribed to the surface enrichment of PMS and pollutants. An estimation of the reactant concentration distribution by molecular dynamic (MD) simulation showed a 1.8-fold higher maximum PMS concentration and 3.9-fold higher 4-CP concentration over the CoNC-MSiI than those of the CoNC-Si (Fig. 3D–E). Such raised local concentrations of pollutant and PMS is supposed to accelerate the surface catalytic reactions accordingly, but could not fully explain the tenfold increased specific activity of the catalyst under nanoconfinement. Therefore, some factors other than reactant enrichment might also contributed to the drastically improved reactivity and PUE of the nanoconfined Co SAC.” (Page. 7)

15. Big issue. The authors are strongly recommended to check their submitted manuscript. All figures should be self-explained. Meanwhile, Fig. 5 was partially displayed. What is the meaning of the X and Y axis titles of Fig. 5I?

Response: We apologize for the incomplete figure display and insufficient explanation. We have thoroughly checked and revised the manuscript to address all these issues. In the original Fig. 5I (now Fig. 3D), the X-axis coordinate represents the distance between the reactant molecules (such as PMS and 4-CP) and the Co atoms on the catalyst surface, and the Y-axis coordinate is a radial distribution function, representing the distribution of reactant concentration at different distances. The above information has been provided in the diagram.

Revision:

Figure 3D. Radial distribution function of PMS distribution concentration of different catalysts obtained by molecular dynamic simulation.

16. In addition, it retained high activity within the pH range of 3-11. Is the PH stable during the AOP process? At least the PH change during the whole reaction period should be given for one group of the results shown in Fig. 2.

Response: Following your nice suggestion, we have conducted extra experiments to evaluate the pH changes in the CoNC-MSi1/PMS system. The results show that, after addition of the acidic PMS (0.4 mM), the solution pH decreased from the initial values of 5.1~9.0 to 3.2~ 3.9 within 5 min. However, the strong acid group (initial pH= 3.2) and strong alkaline group (initial pH= 11.0) showed only slight pH changes. All these test groups, regardless of their actual reaction pH, showed high catalytic activity for pollution degradation (Fig. 6C), confirming the good pH versatility of the nanoconfined catalytic systems. A figure showing the pH changes (including both the initial and final pHs) has been supplemented as suggested (Supplementary Fig. 34). In addition, label in Fig. 6C has been revised as “Initial pH” for a more accurate expression.

Revision:

Figure 6C. 4-CP degradation kinetics under different initial pH values and environmentally-relevant ions. Reaction condition: [catalyst] = 0.25 g·L⁻¹, [PMS] = 0.4 mM, [4-CP] = 0.1 mM, initial solution pH = ~3.2~11.0. [cations] = [anions] = 5 mM.

Supplementary Figure 34. pH change of the CoNC-MSi1 catalytic system after reaction. Reaction condition: [catalyst] = 0.25 g·L⁻¹, [PMS] = 0.4 mM, [4-CP] = 0.1 mM, initial solution pH = ~3.2~11.0.

“..... The 4-CP degradation performance of the CoNC-MSi1/ PMS system was nearly unaffected by diverse environmentally-ubiquitous inorganic ions (Na⁺, K⁺, Ca²⁺, Cl⁻, F⁻, CO₃²⁻, and PO₄³⁻), natural organic matters (NOM) and in broad pH range (Fig. 6C, Supplementary Fig. 32–34).” (Page. 11)

17. How to understand the increased TOC removal with time shown in Supplementary

Fig. 35?

Response: The increased TOC removal over time was mainly due to a self-activation of PMS by the pollutant oxidation intermediates. The HPLC-MS analysis identified the generation of 1,4-benzoquinone (1,4-BQ) intermediate during 4-CP degradation. Benzoquinones are known to serve as PMS activator for enhancing Fenton-like oxidation of pollutants (*Applied Catalysis B: Environmental*. 2023, 320, 121980; *Water Research*. 2017, 125, 209-218). For validation, we evaluated the degradation of 4-CP and its two major intermediates, Ph (phenol) and HQ (hydroquinone), in the CoNC-MSi1/PMS system in the presence of 1,4-BQ. These reaction systems exhibited 1.8~3.0-fold higher degradation rate constants (k_{obs}) than the control (without 1,4-BQ) (Fig. R1a–b), confirming an obvious promotion effect of 1,4-BQ on the pollutant mineralization. Therefore, with the continuous generating of 1,4-BQ and possibly also other quinone intermediates from 4-CP, the 4-CP degradation process was self-accelerated, resulting in increased TOC removal over time.

Revision:

Figure R1. (a) degradation effects of produced 1,4-BQ/PMS systems against PMS only systems for several different intermediates including 4-CP, Ph and HQ, and (b) corresponding kinetic constant (k_{obs}). Reaction condition: [1,4-BQ] = 0.2 g L⁻¹, [PMS] = 2 mM, [pollutants] = 0.05 mM.

18. How about the chemical state of Co in the used catalyst, which is important to show

its stability. As such, XPS analysis should be supplemented for the catalyst after usage, however, which is missing.

Response: Accepting the reviewer's nice suggestion, we have supplemented the XPS analysis for the CoNC-MSi1 after reaction. The Co 2p XPS spectrum of the used catalyst shows no discernable change in the peaks composition and positions, indicating a stable Co valance (Supplementary Fig. 38a). In addition, similar signals of Co, N, C, O, and Si were detected in both the pristine and used catalysts by the full survey scan spectrum (Supplementary Figure 38b). All these evidences confirm a high compositional and structural stability of the CoNC-MSi1 catalyst. The relevant information has been supplemented in the revised manuscript.

Revision:

Supplementary Figure 38. XPS spectrum of (a) Co 2p, and (b) the full survey scan for the pristine and used CoNC-MSi1 catalyst.

Note: The Co 2p XPS spectra showed that the peaks composition and positions were almost unchanged after reaction²⁰. In addition, the peaks of Co, N, C, O, and Si in the full survey scan were also unchanged, confirming a good chemical structure stability of the CoNC-MSi1 catalyst.

“..... Although a slight performance decline occurred during prolonged operation due to catalyst passivation by incompletely degraded intermediates (Supplementary Fig. 35), the activity could be fully recovered by eluting the catalyst with ethanol to remove the surface-accumulated intermediates, indicating a superior structural stability of the

CoNC-MSi1. Its high stability was also supported by its unchanged morphology and structure (Supplementary Fig. 36–38) as well as the negligible Co^{2+} leaching (only 0.007 mg/L) after the decontamination reaction.” (Page. 11–12)

19. Big issue. Fig.6. In terms of application, there will be numerous pollutants including the electron-rich and the electron-deficient. Herein, the ETP mechanism is efficient in the removal of electron-rich pollutants rather than the electron-deficient with a low mineralization rate. However, water decontamination does not only mean pollutant removal and also mineralization. More importantly, toxicity reduction including feed and the generated intermediates should be ensured in terms of water decontamination, which however is missing herein. As such, the shortage of this technique should also be mentioned and the evaluation of toxicity should be supplemented.

Response: Thanks for your constructive and valuable suggestion. This is a very important issue, and we fully agree that selective decontamination does have its limitations. Hence, finding appropriate water treatment niches will be vital for its practical application. **Considering the high efficiency of our catalytic system for selective removal of electron-rich pollutants and superior tolerance to high-concentration inorganic salt, we envisage it may offer an attractive supplement to several other treatment technologies in practical application.** For example, it may serve as a pretreatment, followed by adsorption and biodegradation, to deal with the actual industry wastewater, especially those from refinery and chemical engineering industries (containing high-concentration salt and abundant electron-rich phenolic compounds). In this case, a selective decontamination pretreatment may significantly lower the workload of downstream treatments and enable cost saving in the overall process. In addition, it may also serve as an effective technology for deep purification of secondary effluent from wastewater treatment plants, which contain many electron-rich micropollutants. A combination of selective oxidation with adsorption may not only achieve efficient removal of micropollutants but also prolong the operating time of adsorbent through in-situ degradation of partial surface-accumulated pollutants.

We also added a toxicity assay following the reviewer's suggestion. The 4-CP containing lake water was treated in the continuous-flow reactor, and the toxicity of influent and effluent was assessed by measuring the viability of *E. coli* cells after 6-h exposure (*Sci Rep.* 2021, 11, 15978; *Nat Rev Microbiol.* 2011,9, 779–790) (Supplementary Fig. 42). The bacteria incubated with the influent (4-CP-containing lake water) showed only 15% viability relative to the blank lake water group (cultivated with LB medium), confirming a high acute toxicity of the 4-CP pollutant. In comparison, the effluent-incubated cells still remained > 70% viability. Consistently, the influent-incubated group showed severely suppressed growth while the effluent-incubated group restored growth. All these results confirm **a drastically reduced toxicity of the water sample after treatment in the continuous-flow reactor**. The toxicity reduction was consistent with the efficient 4-CP degradation by the catalytic system, achieving a mineralization degree of 70% and producing some less-toxic intermediates such as maleic acid, propanedioic acid, crotonyl alcohol, and acetic acid.

The above newly-added data and relevant discussions, including the shortage and possible application niches, have been provided in the manuscript.

Revision:

“..... It has to be pointed out that the selective decontamination technologies, like the nanoconfined catalytic system here, mainly target the electron-rich pollutants and are less efficient for removal of the electron-deficient ones (e.g., many halogenated organic pollutants). Thus, to promote its practical application, an appropriate integration with conventional technologies such as adsorption, Fenton oxidation, and biological treatment would be necessary. For example, it may serve as a low-cost pretreatment technology to remove the majority of phenolic compounds in wastewater from refinery and chemical engineering industries, thereby lowering the working load and cost of downstream treatments. Or, it may be combined with an adsorption process to efficiently remove micropollutants from environmental water bodies or wastewater

treatment effluent at minimal chemical consumption. All these possibilities warrant further investigation and demonstration.” (Page. 13)

Supplementary Figure 42. The toxicity assessment of continuous-flow reactor system.

(a) *E. coli* colony after 6-h cultivation in different water samples, (b) relative cell viabilities, and (c) growth curves in different systems.

Note: The toxicity reduction during continuous-flow reactor was proved by the *E. coli* toxicity assessment. The influent lake water contained 4-CP shows about 85% number lethality of the cells, confirming a high acute toxicity of initial lake water (Supplementary Figure 42a-b). In comparison, the effluent-incubated cells still remained >70% viability. Consistently, the growth curves of *E. coli* also showed a severely suppressed growth of the influent-incubated group and a restored growth of the effluent-incubated group (Supplementary Fig. 42c). These results confirm a drastically reduced toxicity of the water sample after treatment in the continuous-flow reactor. The toxicity reduction was consistent with the efficient 4-CP degradation, with

a mineralization degree of 70% achieved, suggesting the potential of practical application in the future.

“..... In addition, the lake water treatment effluent exhibited low biological toxicity and negligible metal ions leaching (Supplementary Fig. 42–43), confirming the high decontamination capability and good environmental benignity of our nanoconfined catalytic system.” (Page. 12)

20. Fig. 6E, how about the particle size of Al_2O_3 ? Will these Al_2O_3 fillers facilitate PMS activation? For the packed-bed reactor, how to make sure there will be no leaching of catalyst and Al_2O_3 fillers?

Response: The average particle size of Al_2O_3 was ~500 nm, slightly larger than the CoNC-MSi1 particles (~400 nm). Our control experiment showed negligible 4-CP removal by Al_2O_3 alone (**without CoNC-MSi1**) (Supplementary Fig. 41b), indicating that **the Al_2O_3 fillers are incapable of PMS activation.**

In addition, according to the leaching tests, the average Co and Al concentrations in the effluent from the packed-bed reactor, with high loading of Al_2O_3 and CoNC-MSi1 and under acidic reaction conditions, were only 0.026 and 0.075 mg/L, respectively (Supplementary Fig. 43). Both values are below the limits of “Guidelines for Drinking-Water Quality, 4th edition” (World Health Organization) and China’s environmental quality standards for surface water (GB3838-2002) (Co: 1.0 mg/L, Al: 0.2 mg/L). The leached amount accounted for 0.006% of the Co content and 0.01% of the Al content in the fillers, confirming the good stability of the filler material. The leaching of Co and Al was even less under the environmental-relevant neutral pH conditions (0.002 and 0.009 mg/L respectively in the effluent) (Fig. R2). Overall, **these results confirm a negligible metal leaching and good stability of our reaction system to facilitate practical water purification application.** Of course, substituting the Al_2O_3 with other fillers may also be considered in practical application to further improve its environmental benignity.

Revision:

Supplementary Figure 41. (b) the 4-CP degradation performance in $\text{Al}_2\text{O}_3/\text{PMS}$ system. Reaction condition: [catalysts] = $0.25 \text{ g} \cdot \text{L}^{-1}$, [PMS] = 0.4 mM, [4-CP] = 0.1 mM.

Supplementary Figure 43. The leaching amounts of Co^{2+} and Al^{3+} during continuous-flow operation for treatment of lake water.

Note: The Co and Al leaching concentration detected in the continuous-flow system were only 0.026 and 0.075 mg/L after 200 h operation, respectively. Both values are below the limits of “Guidelines for Drinking-Water Quality, 4th edition” (World Health Organization) and China’s environmental quality standards for surface water

(GB3838-2002) (Co: 1.0 mg/L, Al: 0.2 mg/L). The leached amount accounted for 0.006% of the Co content and 0.01% of the Al content in the fillers, indicating a strong stability and safety of the catalyst in water-purification applications²¹.

“..... In addition, the lake water treatment effluent exhibited low biological toxicity and negligible metal ions leaching (Supplementary Fig. 42–43), confirming the high decontamination capability and good environmental benignity of our nanoconfined catalytic system.” (Page. 12)

Figure R2. (a) The 4-CP degradation performance and (b) corresponding Co²⁺ and Al³⁺ leaching in the batch experiments before and monitored pH to 7. Reaction condition: [catalysts] = 0.25 g · L⁻¹, [PMS] = 0.4 mM, [4-CP] = 0.1 mM.

Reponses to Reviewer #2's Comments

The work described here doesn't meet the publication standards typically associated with journals like Nature Communications. The three major reasons are i) the material prepared in this study doesn't have any novel aspect. ii) The results on nanoconfinement is not satisfactory based on the results provided, and iii) The chosen problem and its outcomes lack interest and do not offer novel insights in this field. The quality of discussion throughout the manuscript needs an improvement. In some place's authors used the scientific terms that are not defined. Therefore, this manuscript does not meet the required standard. A very few comments that are provided below.

Response: We sincerely appreciate your critical comments and apologize for the unclear expressions in our previous manuscript, which might have caused some confusion and misunderstanding. Herein, we have thoroughly revised the manuscript by adding new evidence, strengthening discussions, and improving the language to better highlight the novelty, scientific contribution, and practical significance of our work. A point-by-point explanation is given below.

i) the material prepared in this study doesn't have any novel aspect.

Response: The novelty of the fabricated material in our work mainly lies in its unique design and performance. The previously reported SACs were constructed typically using carbon material as support, which are difficult to precisely tune the pore sizes and have insufficient mechanical strength for packed-bed reactor operation. Here, we for the first time used the fine-tunable and structurally-stable MSi particles as the support to construct the nanoconfined SAC, which provides an ideal model for the mechanism study and also favors practical application (*Nature Synthesis*. 2022, 1, 658–667; *Nature Communications*. 2022, 13, 295). In addition, we for the first time realized decontamination via highly-selective ETP pathway in SAC-based Fenton-like catalysis, and achieved unprecedented oxidant utilization efficiency (96.6 %) in such systems.

Revision: “..... Here, we demonstrated that, apart from strengthening surface enrichment of the pollutant and oxidant, appropriate nanoconfinement of SACs could also fundamentally alter the Fenton-like catalytic pathway to favor more efficient water decontamination and oxidant utilization. In this proof-of-concept study, we constructed a model catalytic system by confining Co single atoms within nano-sized pores of core-shell-structured mesoporous silica (MSi) particles. Each particle consisted of a spherical silica core and a mesoporous shell (with abundant silica nanopores being covered by a nitrogen-doped carbon layer). The nanopore sizes of the MSi shell were fine-tuned to adjust the nanoconfinement degree. For comparison, a nonporous silica particle (Si) covered by nitrogen-doped carbon layer was set as the unconfined control.”
(Page. 3)

ii) The results on nanoconfinement is not satisfactory based on the results provided.

Response: We appreciate the reviewer’s comment but respectfully disagree with this point. Firstly, nanoconfinement of the SAC resulted in 34.7-fold faster pollutant degradation and drastically raised PMS utilization efficiency (from 61.8% to 96.6%), indicating drastically improved performance. Moreover, the CoNC-MSi1 outperformed almost all the reported heterogeneous catalysts (with primarily nonradical pathways) in specific activity and PMS utilization efficiency for pollutant degradation. The nanoconfinement-induced performance improvement has been highlighted in the revised manuscript.

Revision: “..... Confining the cobalt SAC within the nanopores of mesostructured silica particles fundamentally altered its electronic structure, triggering the transition from singlet oxygen to electron transfer pathway for direct oxidation of 4-chlorophenol. The nanoconfinement-induced pathway change and reactant enrichment collectively led to 34.7-fold increased degradation reactivity of the catalytic system and drastically raised utilization efficiency of peroxymonosulfate (PMS) from 61.8% to 96.6%. It also demonstrated superior activity for degradation of multiple electron-rich phenolic compounds, good environment robustness, and high stability and efficiency for treating

real lake water in a continuous-flow reactor. Our findings expand the knowledge of nanoconfined catalysis and may guide the development of SACs-based materials and technologies towards low-carbon water purification application and beyond.” (Page 2)

iii) The chosen problem and its outcomes lack interest and do not offer novel insights in this field.

Response: We apologize for not clearly describing the novelty and significance of our work, and have thoroughly revised the manuscript to highlight these points.

Firstly, this work aims to address two important bottlenecks of the heterogeneous Fenton-like technologies for recalcitrant pollutant degradation- sluggish interfacial mass transfer and low efficiency of oxidant utilization. Therefore, the chosen problem is of practical importance.

Secondly, we proposed a facile and effective strategy to achieve the above goal: nanoconfinement of Co SAC enabled reactant enrichment and triggered the ETP catalytic pathway, resulting in a 34.7-fold higher degradation rate and drastically raised PMS utilization efficiency from 61.8% to 96.6% for pollutant removal. Achieving simultaneous improvement in activity and oxidant utilization efficiency is very challenging for catalyst optimization. Therefore, the improvement effects are remarkable.

Lastly, we unveiled a fundamental catalytic pathway transition of Co SAC triggered by nanoconfinement- a phenomenon that profoundly affects the catalytic performance but has been ignored in previous studies. Therefore, our findings provide valuable new insights into the nanoconfined effects in heterogeneous catalysis.

Revision:

“..... Over the past few years, a number of heterogeneous catalysts that facilitate nonradical Fenton-like reactions have been developed^{22, 23}. In particular, single-atom catalysts (SACs), being highly reactive and structurally-tunable, were elaborated to enable pollutant removal at drastically increased degradation kinetics and less oxidant input than the radical-based counterpart^{24, 25}. Nevertheless, the limited access to

reactants for the surface active sites and the short diffusion range of the generated active species, dominated by $^1\text{O}_2$ in the existing SAC-based Fenton-like catalytic systems, still considerably restrict the overall efficiencies of decontamination and oxidant utilization^{26, 27}.

The demand for addressing these challenges has motivated the optimization of SAC design, for which engineering strategies such as heteroatom doping and vacancy introduction are commonly adopted^{28, 29, 30}. Recently, attempts have also been made to confine the SAC active sites within a nano-structured space, which was found to drastically raise the local concentration of reactive species and pollutants^{1, 2}, enabling 2~3 order-of-magnitude higher activities for Fenton-like oxidation of pollutants than the unconfined analog^{3, 4}. Nanoconfinement could also alter the physicochemical properties of metal catalytic sites, thereby affecting the thermodynamics of surface-reactant interactions and enhancing or altering the product selectivity and even triggering catalytic pathway shift^{5, 6}. For example, confining the Fe(III)-anchored metal-organic framework nanoparticles within the graphene aerogel substrate fundamentally changed the phenol oxidation pathways from ring-opening degradation to oligomerization route in Fenton-like catalysis³¹. Notably, these studies were focused exclusively on metal nanoparticles. To date, it remains unclear whether catalytic pathway change would also occur for nanoconfined SACs and how such changes would affect the Fenton-like oxidation processes.

Here, we demonstrated that, apart from strengthening surface enrichment of the pollutant and oxidant, appropriate nanoconfinement of SACs could also fundamentally alter the Fenton-like catalytic pathway to favor more efficient water decontamination and oxidant utilization.” (Page. 2–3)

PUE definition: “..... Here, PUE is defined as the ratio of the equivalent amount of PMS (the sole oxidant) (Supplementary Fig. 15) consumed for 4-CP mineralization to the decomposed PMS amount within a given time frame and was estimated according

to the TOC change and PMS consumption in the catalytic systems³² (Supplementary Fig. 16).” (Page. 6)

EtOH, TBA, MB, and FFA definition: “..... EtOH, TBA, MB, and FFA were used to detect $\cdot\text{OH}$, SO_4^- , surface-bound radicals, and $^1\text{O}_2$, respectively.” (Page. 7)

DPBF definition: “..... The concentration of IO2 was analyzed by using 1,3-diphenylisobenzofuran (DPBF) as a probe for spectral detection with a UV-1800 at 410 nm.” (Supplementary information)

1. After reading this manuscript, it invites several fundamental questions. i) Why does authors choose such a complicated synthetic procedure that involves organic solvents from toluene to n-hexane having multiple reactions steps. ii) Even after a hard synthesis procedure, it is disappointing to see a surface area of $50 \text{ m}^2 \text{ g}^{-1}$ maximum for a catalyst. iii) After all, I didn't see any positive aspect in using this catalyst for degradation because it doesn't give any advantage compared to already reported PDS/PMS activation catalysts. The degradation performance showed in this paper are almost identical to some of the carbon materials like activated carbon reported already.

Response: We sincerely appreciate the constructive comments, which are very helpful for guiding the manuscript revision. Herein, we have thoroughly revised the manuscript to highlight our aims, innovations, and contributions. The advantages and limitations of the new catalyst were also discussed. A point-by-point explanation is given below.

i) Why does authors choose such a complicated synthetic procedure that involves organic solvents from toluene to n-hexane having multiple reactions steps.

Response: Elucidation of the nanoconfinement effects and mechanisms in a SAC-based Fenton-like catalytic system requires the construction of a model system with decipherable coordination structure of single atoms and adjustable confinement space. Mesoporous silica (MSi) particles offer an ideal option, which has been employed to immobilize active metals for various confined reactions (*Chem Catalysis*. 2022, 2,

1893-1918). In addition, the high mechanical strength of MSi particles also favors their application in pressurized reactors for water treatment. To enable precise control of the nanopore sizes of MSi, we adopted a synthetic procedure following the literature reports (*J. Am. Chem. Soc.* 2015, 137, 41, 13282–13289; *Chem.* 2021, 7, 1020-1032; *Nano Lett.* 2014, 14, 2, 923–932).

We want to stress that this work aims to showcase a new material design with well-controlled material properties. That's why we adopted a relatively complicated material synthesis method. For promoting practical application, in future studies, we can further optimize the synthesis procedures and even use other support materials to make the catalyst synthesis process more simple, cost-effective, and scalable.

ii) Even after a hard synthesis procedure, it is disappointing to see a surface area of 50 $\text{m}^2 \text{g}^{-1}$ maximum for a catalyst.

Response: As mentioned above, this is a proof-of-concept study that shed new insights into the nanoconfinement effects, and provided a new strategy to address the interfacial mass transfer and oxidant utilization efficiency issues of SAC-based Fenton-like catalysis for water treatment. Thus, we mainly focused on regulating the pore sizes of the catalyst that are closely related to the nanoconfinement effect, while providing no optimization of the other material properties such as specific surface area. Increasing the specific area of the catalyst material may be easily achieved by means such as reducing the particle size, increasing the shell thickness, and adopting more porous substrate. We prefer to do such material optimization in our future studies for further promoting practical application.

In addition, we have re-measured the specific area of the catalyst and found the actual area was 85.89 $\text{m}^2 \text{g}^{-1}$ instead of 50 $\text{m}^2 \text{g}^{-1}$ (Fig. 1C and Supplementary Table 1). Although this value was still low, our catalyst already achieved superior activity surpassing most of the state-of-the-art heterogeneous catalysts in spite of their much larger specific surface areas (Supplementary Table 6). Furthermore, it has to be pointed out that a higher surface area does not necessarily translate to an improved performance

in applications, especially for some diffusion-limited processes (*Nat Rev Mater.* 2016, 1, 1602). Therefore, a systematic optimization of various material properties in the future will be highly desired to further improve the catalyst performance.

Revision:

Figure. 1C. N₂ adsorption-desorption isotherms and the pore size distribution curves in the illustration of two CoNC-MSi and CoNC-Si catalysts.

Supplementary Table 1. BET surface area, pore diameter, and pore volume of the prepared samples.

Catalyst	BET surface area (m ² g ⁻¹)	Pore diameter (nm)	Pore volume (cm ³ g ⁻¹)
CoNC-MSi1	85.89	7.71	0.15
CoNC-MSi2	74.27	12.83	0.21
CoNC-Si	14.48	/	/

iii) After all, I didn't see any positive aspect in using this catalyst for degradation because it doesn't give any advantage compared to already reported PDS/PMS activation catalysts.

Response: This catalyst in its current form (without sufficient optimization) already exhibited many remarkable advantages over the existing ones in Fenton-like catalysis,

including ultra-high decontamination activity and PMS utilization efficiency, (Fig. 3A-B) surpassing those of almost all the existing nonradical Fenton-like catalytic systems (Fig. 3C, Supplementary Table 6). In particular, the ETP-predominated pathway endowed our catalytic system with extraordinary selectivity and environmental robustness for degradation of electron-rich organic pollutants at minimal chemical input (Fig. 6A-B). This has been demonstrated by our 200-h continuous-flow experiment for treatment of contaminated lake water (Fig. 6F).

Therefore, the novel catalytic system may be used as a low-carbon supplementary technology for pretreatment of high-salinity industrial wastewater and for deep purification of WWTP secondary effluent, which are difficult for the existing advanced oxidation processes. Besides that, in light of its mediated ETP pathway, the nanoconfined catalytic system may also enable spatially decoupled pollutant oxidation and oxidant activation processes (i.e., without contact between the oxidant and the water pollutants) to facilitate aquatic environmental remediation, thereby avoiding the wasting and ecological risks of conventional advanced oxidation systems.

A detailed discussion about the advantages, current limitations, and future opportunities of the catalytic system has been added to the manuscript.

Revision:

“..... In this work, we discovered a fundamental catalytic pathway transition of Co SAC triggered by nanoconfinement- a phenomenon that has been ignored in previous nanoconfined SAC studies. In particular, a fine modulation of the catalytic pathway within the nonradical catalytic regime was enabled for the first time, and underlying mechanisms were clarified. These findings significantly broaden our knowledge of nanoconfinement catalysis, which has been focused mainly on nanoscale catalysts previously⁵. Importantly, by leveraging the nanoconfinement effect to fine-tune the local electronic structure and chemical properties of the metal sites, highly efficient SAC could be constructed. The CoNC-MSi1/PMS system here exhibited unprecedentedly high activity and PUE for selective degradation of electron-rich pollutants, and demonstrated superior stability and environmental robustness, making it an attractive

option for low-carbon wastewater treatment. In particular, considering the superior mechanical strength of the CoNC-MSi1 catalysts, it might be potentially incorporated into membrane design to facilitate up-scaled membrane catalytic decontamination applications. In addition, with a mediated ETP pathway, it may offer opportunities for the development of spatially decoupled pollutant oxidation and oxidant activation processes (i.e., without contact between the oxidant and the water pollutants) to facilitate aquatic environmental remediation³³, avoiding the wasting and ecological risks of conventional advanced oxidation systems. Of course, the nanoconfinement strategy demonstrated here could also be readily employed for construction of various other SACs with lower costs and improved scalability, thus better adapting such nanoconfined systems to practical application niches.

It has to be pointed out that the selective decontamination technologies, like the nanoconfined catalytic system here, mainly target the electron-rich pollutants and are less efficient for removal of the electron-deficient ones (e.g., many halogenated organic pollutants). Thus, to promote its practical application, an appropriate integration with conventional technologies such as adsorption, Fenton oxidation, and biological treatment would be necessary. For example, it may serve as a low-cost pretreatment technology to remove the majority of phenolic compounds in wastewater from refinery and chemical engineering industries, thereby lowering the working load and cost of downstream treatments. Or, it may be combined with an adsorption process to efficiently remove micropollutants from environmental water bodies or wastewater treatment effluent at minimal chemical consumption. All these possibilities warrant further investigation and demonstration.” (Page. 13)

2. By looking at the particle size of MSN (300 nm), CoNC-MSN1 (400 nm), and CoNC-MSN2(400 nm) are they really nanoparticles? Throughout the manuscript this terminology radiates.

Response: Thanks for pointing out this issue. Indeed, it is inappropriate to name our catalysts (300-400 nm particle sizes) as “nanoparticles”. Therefore, following the

terminology used in other studies (*Nano Lett.* 2020, 20, 5, 4014–4021), we have renamed the nanoconfined materials as mesoporous silica particles (MSi) and the unconfined control as silica particles (Si).

3. Under what conditions CoNC-MSN1 (400 nm), and CoNC-MSN2 (400 nm) are achieved. It is not mentioned in the material synthesis section.

Response: Accepting your suggestion, we have added in the Method section a detailed description of the material synthesis methods. “To tune the pore size of the particles, the polarity of the upper organic solution was regulated by controlling the ratio of n-hexane and toluene as 1:0 to obtain CoNC-MSi1 (with a pore size of 7.7 nm), and 1:1 to produce CoNC-MSi2 (with a pore size of 12.8 nm), respectively.”

Revision:

Supplementary Figure 1. Operating procedures for synthesis of the nanoconfined and unconfined CoNC catalysts.

4. First of all, in page.3, TEM analysis is not suitable for accurately measuring the pore size of any material. This sentence needs to be revised.

Response: Accepting the reviewer’s suggestion, we have provided the N₂ adsorption-

desorption isotherms to more accurately reveal the pore size distribution information (Fig. 1C), and rephrased the sentence accordingly.

Revision:

Figure. 1C. N₂ adsorption-desorption isotherms and the pore size distribution curves in the illustration of two CoNC-MSi and CoNC-Si catalysts.

“..... According to the scanning electron microscopy (SEM) and transmission electron microscopic (TEM) images, the nanoconfined catalysts had relatively larger particle sizes than CoNC-Si (~400 vs. ~310 nm) due to the presence of a porous shell (Fig. 1A–B, Supplementary Fig. 2). The N₂ adsorption-desorption isotherms and the pore size distribution curves reveal different pore sizes of the two nanoconfined catalysts (Fig. 1C, Supplementary Table 1). The catalysts with average pore sizes of 7.7 nm and 12.8 nm were referred to as CoNC-MSi1 and CoNC-MSi2, respectively (Supplementary Table 1). The relatively small pore sizes of the CoNC-MSi1 were in line with its more densely-distributed nanopores in the shell, as manifested by the TEM mages (Fig. 1B, Supplementary Fig. 2).” (Page. 3)

5. The authors assert an average nanopore size within the range of 7.7 to 12.8 nm. To our surprise, no mesoporous features were noticed in the BET data. As this pore size falls within the mesoporous scale type-IV isotherm is missing why?

Response: Thanks for your valuable comments. To address your concern, we have

repeated the BET measurements (Fig. 1C). The updated N₂ adsorption–desorption isotherms of two CoNC-MSis clearly showed typical type IV isotherms with a type H2 hysteresis loop according to Brunauer–Deming–Deming–Teller (BDDT) classification, indicating a predominantly mesoporous structure of the CoNC-MSi materials. In contrast, the N₂ adsorption–desorption isotherms of CoNC-Si catalyst showed typical type III isotherms, corresponding to its nonporous structure.

Revision:

Figure. 1C. N₂ adsorption–desorption isotherms and the pore size distribution curves in the illustration of two CoNC-MSi and CoNC-Si catalysts.

6. In page. 3 what does the 10¹⁰-times larger surface area mean. There is no such an increase in BET results (Table S1).

Response: Thank you for pointing out this issue. The statement of “10¹⁰-times larger specific surface area” was made based on a theoretical estimation of the specific surface area of an ideal MSi sphere, which is indeed inaccurate. The BET analysis can provide more reliable information about specific surface area. Therefore, we have re-measured the N₂ adsorption–desorption curves, which revealed 6.8-fold larger specific surface area of the CoNC-MSi1 than the nonporous counterpart (Fig. 1C). Therefore, we have corrected the statement in the revised manuscript.

Revision:

“..... The N₂ adsorption-desorption isotherms and the pore size distribution curves reveal different pore sizes of the two nanoconfined catalysts (Fig. 1C, Supplementary Table 1). The catalysts with average pore sizes of 7.7 nm and 12.8 nm were referred to as CoNC-MSi1 and CoNC-MSi2, respectively (Supplementary Table 1). The relatively small pore sizes of the CoNC-MSi1 were in line with its more densely-distributed nanopores in the shell, as manifested by the TEM images (Fig. 1B, Supplementary Fig. 2). The porous structure also endowed the CoNC-MSi1 with 6.8-fold larger specific surface area than that of the nonporous CoNC-Si (Supplementary Table 1).” (Page. 3)

7. In page.3 The statement “stabilization of Co single atom by the Si-OH groups” is contradictory with “a typical feature of Co SAC with N coordination” in the next paragraph. The claims made should be consistent throughout the manuscript.

Response: We apologize for the inconsistent statement. In our catalyst, the Co atoms were coordinated with N instead of Si, while the Si-OH group in the silica support plays a role in promoting the formation of carbon layers during calcination (*Applied Catalysis B: Environmental*. 2015, 176–177, 212-224). The statement “stabilization of Co single atom by the Si-OH groups” has been deleted.

Revision:

“..... As manifested by the aberration-corrected high-angle annular dark-field scanning transmission electron microscopy (AC-HAADF-STEM) images (Fig. 1E) and the elemental mapping (Fig. 1D, Supplementary Fig. 3), atomically dispersed bright spots, with homogeneously distributed Co, N, and C elements, were identified in all the catalysts³⁴. This result suggests that the Co single atoms were anchored within the N-doped carbon layer on the surface of the silica skeleton³⁵. The presence of Co element predominantly in the form of single atoms was also supported by the X-ray diffraction (XRD), Raman, and FT-IR spectra, which showed no obvious signal of Co-Co bond in all the catalysts³⁶ (Supplementary Fig. 4–6).” (Page. 3)

8. If N-atoms from carbon are the coordination sites of Co single-atoms. What is the

role of SiO₂ here. Why it is not removed through etching?

Response: As mentioned above, the SiO₂ provides Si-OH groups to immobilize the nitrogen-doped carbon layers, and the latter offers coordination sites for Co single-atoms. Such a configuration could reduce the agglomeration of Co atoms during calcination (*Nat Commun.* 2022, 13, 295; *ACS Nano.* 2023, 17, 5, 5025–5032). Therefore, the SiO₂ played an important role in maintaining the catalyst material structure, and hence could not be etched.

Besides, the SiO₂ core endowed the catalyst with superior mechanical strength to enable pressurized operation, which otherwise would lead to structure collapse. This was demonstrated by its good water treatment performance in a continuous-flow compacted-bed reactor under ~10 MPa pressure (Fig. 6F). Therefore, the SiO₂ core is important for the catalyst application. The relevant information has been provided in the manuscript.

Revision:

“.....This result suggests that the Co single atoms were anchored within the N-doped carbon layer on the surface of the silica skeleton³⁵. The presence of Co element predominantly in the form of single atoms was also supported by the X-ray diffraction (XRD), Raman, and FT-IR spectra, which showed no obvious signal of Co-Co bond in all the catalysts³⁶ (Supplementary Fig. 4–6).” (Page. 3)

“.....The superior mechanical strength of the catalyst (with a silica core) makes it adaptable to pressurized operation. During the 200-hour continuous operation under ~10 MPa pressure, the system steadily achieved 100% 4-CP removal and 70% TOC removal from the lake water at a short hydraulic retention time of only 3.24 min (Fig. 6F, Supplementary Fig. 41), confirming a high decontamination efficiency and stability.” (Page. 12)

9. It is apparent that results of ¹O₂ suppression from CoNC-SN to CoNC-MSN2, and CoNC-MSN1 is well correlated with Co concentration in the system, rather than the nanoconfinement effect claimed by authors. The kobs in Fig. 3B agrees well with it.

Response: Thank you for your insightful comments. It is true that the $^1\text{O}_2$ suppression (corresponding to the proportion of $^1\text{O}_2$ pathway) was the lowest for CoNC-MSi1, which simultaneously had the highest Co loading and nanoconfinement degree among the three tested catalysts. Notably, **there is no contradiction in the impacts of Co loading and nanoconfinement, which were interrelated and were both important reasons for the pathway transition.** So, we guess that the question here is what governs the pathway transition in the catalytic system, Co content or nanoconfinement?

One possibility is that the $^1\text{O}_2$ generation is mainly catalyzed by the Co-free support while the Co atoms primarily contribute to the ETP pathway. In this case, the Co content would play a pivotal role in the pathway transition, as was speculated by the reviewer. However, this possibility was not fully supported by our experimental evidence. Firstly, according to the DPBF probe test results, fully eliminating the Co atoms from the catalyst (i.e., the C-MSi1 group) resulted in slightly decreased $^1\text{O}_2$ yield while drastically increased another pathway (presumably PMS* generation) relative to the CoNC-MSi1 group (Fig. 3I, Supplementary Fig. 25). The considerable generation of PMS* in the Co-free group suggests that Co is not the key to ETP pathway. Instead, the presence of Co even contributed considerably to the $^1\text{O}_2$ generation, as manifested by the obviously improved $^1\text{O}_2$ yield of the Co SACs than those of the Co-free controls. Notably, the CoNC-MSi2 and CoNC-Si showed similar $^1\text{O}_2$ yields, in spite of their different Co contents and nanoconfinement status (Fig. 3I). Altogether, these results indicate that Co-free support played a decisive role in catalyzing the $^1\text{O}_2$ generation, and this pathway was strengthened in the presence of Co single atoms.

As for the ETP pathway, the PMS* yield can be approximated according to the discrepancy between the PMS consumption and the $^1\text{O}_2$ yield, since these were the two major ROS in our catalytic systems. We found that the PMS* yield was increased markedly with raised Co content from 0.01 wt% (CoNC-Si) to 0.05 wt% (CoNC-MSi2), but showed no obvious change when further increasing to 0.07 wt% Co (CoNC-MSi1) (Fig. 3I). Apparently, there was no straightforward correlation between the Co content and $^1\text{O}_2$ / ETP pathway. Therefore, **it was the enforcement of nanoconfinement and**

the presence of Co SAC that synergistically promoted the PMS* generation and triggered the pathway transition in Fenton-like catalysis.

Nevertheless, we want to point out that, although it is difficult to distinguish the impacts of nanoconfinement and Co content due to their close intercorrelation, we think nanoconfinement rather than Co content should be a primary driver for the pathway transition of Co SAC. Theoretically, Co content can alter the reaction kinetics but typically exerts minimal impact on the reaction thermodynamics and the reaction pathway (*Nat Commun.* 2022, 13, 3063). In contrast, the nanoconfinement effect can not only modify the intrinsic activity of active sites but also their interfacial interaction with reactants, potentially altering the reaction's thermodynamic behavior (*Nat Commun.* 2024, 15, 917; *Angew. Chem.* 2022, 134, e202200755).

Overall, nanoconfinement of Co SAC critically affects the catalytic activity and pathway, but the size-dependent nanoconfinement effects on the SAC properties and catalytic behaviors are complicated and warrant further investigation. To address the reviewer's concern, we have added a relevant discussion in the manuscript.

Revision:

Figure 3I. The amount of PMS consumption for 4-CP degradation and the ¹O₂ yield (estimated according to the DPBF conversion). Reaction condition: [catalyst]= 0.25 g · L⁻¹, [PMS] = 0.4 mM, [4-CP] = 0.1 mM, [DPBF] = 0.4 mM.

Supplementary Figure 25. (a) The PMS consumption and DPBF conversion by $^1\text{O}_2$ in three Co-free catalysts, and (b) the relationship between the $^1\text{O}_2$ percentage, (c) the relationship between the $^1\text{O}_2$ percentage and different Co concentration. Reaction condition: [catalyst] = $0.25 \text{ g} \cdot \text{L}^{-1}$, [PMS] = 0.4 mM, [4-CP] = 0.1 mM, [DPBF] = 0.4 mM.

“..... Nanoconfinement can fundamentally alter the coordination environment and many electronic properties of the confined molecules³⁷, potentially altering the reaction's thermodynamic behavior³⁸. Thus, it is interesting to know what material properties of the Co SAC have driven the catalytic pathway change in our system. The mass content, coordination number, and the nanoconfinement effect of the Co active sites were then thoroughly analyzed. Apparently, there was no straightforward correlation between the Co content and $^1\text{O}_2$ /ETP pathway, eliminating its main role in governing the catalytic pathway change (Supplementary Fig. 25).” (Page. 9)

10. The quenching effect observed with FFA is debatable in PMS activation system (<https://doi.org/10.1016/j.chemosphere.2023.138264>). Also, the inhibition can be caused by FFA oxidation itself sometimes.

Response: Thanks for the valuable suggestion. Yes, the quenching effect of FFA in PMS activation system might be biased due to possible FFA oxidation itself by PMS. Therefore, we have added the D_2O experiment and EPR test to ascertain the $^1\text{O}_2$ generation and used DPBF probe to further quantify the $^1\text{O}_2$ yield. (*Environ. Sci. Technol.* 2022, 56, 12, 8833–8843; *Nat Commun.* 2020, 11, 1735.). The multiple evidences strongly confirm the generation of $^1\text{O}_2$ in our catalytic systems. The relevant

descriptions have been supplemented in the revised manuscript as shown below.

Revision:

“..... According to the results of ROS quenching experiments³⁹ (Fig. 3F, Supplementary Fig. 17–18), the pollutant degradation in the CoNC-Si/PMS system was severely suppressed by adding furfuryl alcohol (FFA) as a singlet oxygen (¹O₂) scavenger⁴⁰. To mitigate the inhibition attributed to FFA oxidation itself, the predominant generation of ¹O₂ in this system was also supported by the D₂O experiments and electron spin resonance (ESR) analysis with a TEMP probe⁴¹ (Fig. 3H, Supplementary Fig. 17).” (Page. 7–8)

11. Where is DPBF in Fig. 3D?

Response: DPBF is a chemical probe for ¹O₂ detection and was used for quantification of the ¹O₂ yield here (now Fig. 3I). This information has been added in the figure caption.

Revision:

Figure 3I. The amount of PMS consumption for 4-CP degradation and the ¹O₂ yield (estimated according to the DPBF conversion). Reaction condition: [catalyst]= 0.25 g · L⁻¹, [PMS] = 0.4 mM, [4-CP] = 0.1 mM, [DPBF] = 0.4 mM.

12. Fig. 5 appears incomplete as only half of it is visible. Authors are advised to exercise caution during the uploading and approval of submissions to ensure the figures are fully

visible.

Response: We apologize for not providing the fully-displayed figure. We have thoroughly checked all the figures to avoid such problems.

Figure 5. A–C The charge density difference for PMS adsorption on different catalysts (the light yellow and light blue denote the electron depletion and accumulation, respectively). D Models and affinity of PMS and 4-CP binding on different catalysts. E–F In situ FT-IR spectra of PMS and 4-CP with different catalysts. G Zeta potential of the different catalysts. H Pathway of ¹O₂ generation over different catalysts via PMS activation.

13. Many important publications in this field are missing. For eg. 10.1038/s41467-022-30560-9,

Response: Following the reviewer's nice suggestion, we have updated the cited references. Several other important works in this field published recently, including the one mentioned by the reviewer, have been cited.

Revision:

2. Zhang S, Zheng H, Tratnyek PG. *Advanced redox processes for sustainable water treatment. Nature Water* **1**, 666-681 (2023).
3. Peydayesh M, Mezzenga R. *Protein nanofibrils for next generation sustainable water purification. Nat Commun* **12**, 3248 (2021).
4. Li J, et al. *Mesoporous bimetallic Fe/Co as highly active heterogeneous Fenton catalyst for the degradation of tetracycline hydrochlorides. Scientific Reports* **9**, 15820 (2019).
5. Liang Z, et al. *Effective green treatment of sewage sludge from Fenton reactions: Utilizing MoS₂ for sustainable resource recovery. Proceedings of the National Academy of Sciences* **121**, e2317394121 (2024).
6. Zhang Y-J, et al. *Simultaneous nanocatalytic surface activation of pollutants and oxidants for highly efficient water decontamination. Nat Commun* **13**, 3005 (2022).
7. Thomas N, Dionysiou DD, Pillai SC. *Heterogeneous Fenton catalysts: A review of recent advances. Journal of Hazardous Materials* **404**, 124082 (2021).
9. Jiang J, et al. *Nitrogen Vacancy-Modulated Peroxymonosulfate Nonradical Activation for Organic Contaminant Removal via High-Valent Cobalt-Oxo Species. Environmental Science & Technology* **56**, 5611-5619 (2022).
13. Xin S, et al. *Electron Delocalization Realizes Speedy Fenton-Like Catalysis over a High-Loading and Low-Valence Zinc Single-Atom Catalyst. Advanced Science* **10**, 2304088 (2023).
15. Wang Y, Lin Y, He S, Wu S, Yang C. *Singlet oxygen: Properties, generation, detection, and environmental applications. Journal of Hazardous Materials* **461**, 132538 (2024).
17. Wu L, et al. *Oxygen Vacancy-Induced Nonradical Degradation of Organics: Critical Trigger of Oxygen (O₂) in the Fe–Co LDH/Peroxymonosulfate System. Environmental Science & Technology* **55**, 15400-15411 (2021).
22. Grommet AB, Feller M, Klajn R. *Chemical reactivity under nanoconfinement.*

- Nature Nanotechnology* **15**, 256-271 (2020).
23. Yang Z, Qian J, Yu A, Pan B. Singlet oxygen mediated iron-based Fenton-like catalysis under nanoconfinement. *Proceedings of the National Academy of Sciences* **116**, 6659-6664 (2019).
 24. Liu C, et al. Nanoconfinement Engineering over Hollow Multi-Shell Structured Copper towards Efficient Electrocatalytical C–C coupling. *Angewandte Chemie International Edition* **61**, e202113498 (2022).
 25. Zhang X, et al. Nanoconfinement-triggered oligomerization pathway for efficient removal of phenolic pollutants via a Fenton-like reaction. *Nat Commun* **15**, 917 (2024).
 27. Sheng X, et al. N-doped ordered mesoporous carbons prepared by a two-step nanocasting strategy as highly active and selective electrocatalysts for the reduction of O₂ to H₂O₂. *Applied Catalysis B-Environmental* **176**, 212-224 (2015).
 28. Kumar P, et al. High-Density Cobalt Single-Atom Catalysts for Enhanced Oxygen Evolution Reaction. *Journal of the American Chemical Society* **145**, 8052-8063 (2023).
 32. Chen S, et al. Identification of the Highly Active Co–N₄ Coordination Motif for Selective Oxygen Reduction to Hydrogen Peroxide. *Journal of the American Chemical Society* **144**, 14505-14516 (2022).
 34. Gu C-H, Wang S, Zhang A-Y, Liu C, Jiang J, Yu H-Q. Slow-release synthesis of Cu single-atom catalysts with the optimized geometric structure and density of state distribution for Fenton-like catalysis. *Proceedings of the National Academy of Sciences* **120**, e2311585120 (2023).
 37. Wei Y, et al. Ultrahigh Peroxymonosulfate Utilization Efficiency over CuO Nanosheets via Heterogeneous Cu(III) Formation and Preferential Electron Transfer during Degradation of Phenols. *Environmental Science & Technology* **56**, 8984-8992 (2022).
 38. Guo Z-Y, et al. Crystallinity engineering for overcoming the activity–stability

- tradeoff of spinel oxide in Fenton-like catalysis. Proceedings of the National Academy of Sciences* **120**, e2220608120 (2023).
40. Zhao Y, et al. Janus electrocatalytic flow-through membrane enables highly selective singlet oxygen production. *Nature Communications* **11**, 6228 (2020).
42. Yao Y, et al. Rational Regulation of Co–N–C Coordination for High-Efficiency Generation of $^1\text{O}_2$ toward Nearly 100% Selective Degradation of Organic Pollutants. *Environmental Science & Technology* **56**, 8833-8843 (2022).
53. Zang Y, et al. Selective CO_2 Electroreduction to Ethanol over a Carbon-Coated CuOx Catalyst. *Angewandte Chemie International Edition* **61**, e202209629 (2022).
58. Zhou T, et al. Nanopore Confinement of Electrocatalysts Optimizing Triple Transport for an Ultrahigh-Power-Density Zinc–Air Fuel Cell with Robust Stability. *Advanced Materials* **32**, 2003251 (2020).
60. Wu B, Li ZL, Zu YX, Lai B, Wang AJ. Polar electric field-modulated peroxymonosulfate selective activation for removal of organic contaminants via non-radical electron transfer process. *Water Research* **246**, (2023).
62. Yu Z, et al. Decoupled oxidation process enabled by atomically dispersed copper electrodes for in-situ chemical water treatment. *Nature Communications* **15**, 1186 (2024).
68. Kresse G, Furthmüller J. Efficient iterative schemes for *ab initio* total-energy calculations using a plane-wave basis set. *Physical Review B* **54**, 11169-11186 (1996).
69. Perdew JP, Burke K, Ernzerhof M. Generalized Gradient Approximation Made Simple. *Physical Review Letters* **77**, 3865-3868 (1996).
70. Entwistle MT, Hodgson MJP, Wetherell J, Longstaff B, Ramsden JD, Godby RW. Local density approximations from finite systems. *Physical Review B* **94**, 205134 (2016).
71. Sheppard D, Terrell R, Henkelman G. Optimization methods for finding

minimum energy paths. The Journal of Chemical Physics **128**, (2008).

14. In page. 7 what is mean by “as evidenced by its high NaClO₄ resistance”?

Response: NaClO₄ is commonly used to distinguish the surface binding mode of molecules. Less interference by high-concentration NaClO₄ means strong adsorption of the target molecule via inner-sphere interaction (typically via chemical bonds) (*Proceedings of the National Academy of Sciences. 2024, 121, 3*). To address the reviewer’s concern, we have added a brief explanation in the manuscript.

Revision:

“..... In addition, the mediated ETP pathway was also enabled by the inner-sphere binding mode of both the pollutant and PMS (evidenced by the high resistance to NaClO₄ interference as shown in Supplementary Fig. 23)⁴², and by the high conductivity of the nanoconfined catalyst (electric resistance ~480 Ω, graphitic degree $I_D/I_G = 0.86$)⁴³ (Supplementary Fig. 5 and 24).” (Page. 8)

Supplementary Figure 23. 4-CP degradation of the catalytic systems in the presence of NaClO₄. Reaction condition: [catalyst] = 0.25 g · L⁻¹, [PMS] = 0.4 mM, [4-CP] = 0.1 mM, [NaClO₄] = 0 ~ 200 mM.

Note: The increase of ionic strength could profoundly affect outer-sphere interactions (electrostatic bonding) between the solute and the material surface in both equilibrium and kinetics but not affect inner-sphere complexation (covalent bonding or a combination of covalent and ionic bonding)⁴⁴. Therefore, the binding mode can be identified by using NaClO₄ to adjust the ionic strength. We found that 4-CP degradation was almost unaffected under high-concentration NaClO₄, indicating an inner-sphere interaction between the PMS and the CoNC-MSi1 catalyst.

15. Is this the electrical conductivity measuring device? The authors didn't used the work electrochemical impedance spectroscopy.

Response: We apologize for the unclear description. The mediated electron transfer pathway was validated by using a dual-chamber galvanic cell (Fig. 4DE), not an electrical conductivity measuring device. We measured the electric conductivity of the catalysts using electrochemical impedance spectroscopy (Supplementary Fig. 24).

Revision:

Supplementary Figure 24. EIS plots of different catalysts.

“..... To elucidate the specific catalyst-pollutant-PMS interactions in the nanoconfined catalytic system, we constructed a galvanic cell, with the two electrolyte chambers (containing 4-CP and PMS respectively) being separated by a catalyst-loaded carbon flake. This system showed distinct current changes upon 4-CP and PMS addition

accompanied by their synchronous decomposition in individual chambers (Fig. 4D–E, Supplementary Fig. 21), suggesting the existence of mediated ETP pathway for pollutant degradation.” (Page. 8)

There are many such issues in this manuscript that can be pointed out in every paragraph. I recommend authors should draft the manuscript carefully before submitting to any journal.

Response: Your valuable comments are greatly appreciated. We have thoroughly revised the manuscript to improve its readability and provide more detailed explanations. In addition, we double-checked the manuscript to ensure that all supplementary images are mentioned and referred to in the main text.

Specific comments:

In Page.3 There is a typo in (Fig. 1E-F)

Response: Corrected. Thanks

In Page.4 There is a typo in “tothe”

Response: Corrected.

The rate constant symbol kobs, is used in different ways Kobs (Fig. 3), kobs (page.5)

Response: The lower-case k has been uniformly used for all the rate constant symbols.

Define PUE in the manuscript.

Response: The definition of PUE has been added as suggested: *“PUE is defined as the ratio of the equivalent amount of PMS (the sole oxidant) (Supplementary Fig. 15) consumed for 4-CP mineralization to the decomposed PMS amount within a given time frame and was estimated according to the TOC change and PMS consumption in the catalytic systems³² (Supplementary Fig. 16).” (Page. 6)*

In page.7 there is a typo 'peak at ~835 cm⁻'

Response: Corrected. Thanks.

Reponses to Reviewer #3's Comments

This work reports an interesting nanoconfinement effect that has been overlooked in previous studies. Nanoconfined reactions are actually widespread in pollution control systems that applying porous nanomaterials and deserve attention. Although there are already several studies about application nanoconfined single atom catalysts in Fenton-like oxidation, this might be the first report of nanoconfinement-triggered catalytic pathway transition occurring in single atom catalysis system. Especially, while pathway change from radical to nonradical reactions have been reported in Fenton-like systems, a transition between different nonradical pathways is scarce. It also provides new insights to benefit a better understanding of the boosted reaction kinetics in nanoconfined catalysis, which has been widely believed to be primarily ascribed to accelerated interfacial mass transfer. The good performances of the catalytic system with fine-tuned nanoconfinement degree for selective decontamination and stable wastewater treatment operation were demonstrated. The major conclusions are mostly well supported by experimental evidences and theoretical calculations. Overall, I think the current work is of scientific value and practical significance to boost the development of Fenton-like oxidation technologies toward practical application. However, one figure is incompletely displayed and it would be helpful for the readers to better understand this work by clarifying the following issues with a minor revision.

Response: We appreciate the reviewer's positive comments and constructive suggestions, which are very helpful for further improving the quality of our manuscript. A thorough revision of the manuscript has been made (detailed below).

Figure 5. A–C The charge density difference for PMS adsorption on different catalysts (the light yellow and light blue denote the electron depletion and accumulation, respectively). D Models and affinity of PMS and 4-CP binding on different catalysts. E–F In situ FT-IR spectra of PMS and 4-CP with different catalysts. G Zeta potential of the different catalysts. H Pathway of $^1\text{O}_2$ generation over different catalysts via PMS activation.

1. What are the specific advantages of the catalytic pathway transition from singlet oxygen to the electron-transfer process? How is it related to the saving in oxidant consumption? The benefits of such pathway transition in practical decontamination applications should be clarified in more details.

Response: Thanks for the constructive suggestions. We have a more detailed discussion and explanation to clarify these issues and to highlight the scientific and practical significance of our work, in particular the catalytic pathway transition.

Revision:

“..... Therefore, a $^1\text{O}_2$ -to-ETP pathway transition in Fenton-like catalysis was triggered by the nanoconfinement of Co SACs, contributing considerably to the high decontamination activity and PUE of the catalytic systems. Specifically, a considerable fraction of the generated $^1\text{O}_2$ in the unconfined system, due to lack of sufficient contact with pollutants, was self-decomposed during diffusion into the bulk solution⁴⁵, resulting in low PUE. In contrast, the nanoconfined system could not only accelerate the PMS activation and pollutant oxidation due to their raised local concentrations, but also provide long-lasting PMS* to favor more efficient PMS utilization for pollutant degradation (Fig. 4G).” (Page. 8)

“..... The CoNC-MSi1/PMS system here exhibited unprecedentedly high activity and PUE for selective degradation of electron-rich pollutants, and demonstrated superior stability and environmental robustness, making it an attractive option for low-carbon wastewater treatment. In particular, considering the superior mechanical strength of the CoNC-MSi1 catalysts, it might be potentially incorporated into membrane design to facilitate up-scaled membrane catalytic decontamination applications. In addition, with a mediated ETP pathway, it may offer opportunities for the development of spatially decoupled pollutant oxidation and oxidant activation processes (i.e., without contact between the oxidant and the water pollutants) to facilitate aquatic environmental remediation³³, avoiding the wasting and ecological risks of conventional advanced oxidation systems. Of course, the nanoconfinement strategy demonstrated here could also be readily employed for construction of various other SACs with lower costs and improved scalability, thus better adapting such nanoconfined systems to practical application niches.” (Page. 13)

2. In the 4-CP degradation reaction, the performances of catalysts were compared based on different dosages (mass concentration or Co content basis in different cases). Why not using unified dosage? This inconsistency should be addressed or explained.

Response: Thanks for the valuable suggestions. We had deleted the Co content-based comparison and adopted the unified mass concentration for the activity evaluation

following your suggestion (Fig. 3A).

Revision:

Figure 3A. 4-CP degradation performances of different catalyst/ PMS systems.

“..... Attributed to the increased Co loading and altered coordination structure, the nanoconfined Co SACs exhibited drastically different Fenton-like oxidation behaviors from those of the unconfined control. A complete removal of 4-CP within 2 min was achieved in the two nanoconfined catalytic systems, against only ~20% removal by the unconfined analog (Fig. 3A, Supplementary Fig. 9–10).” (Page. 5)

3. To substantiate the advantages of the nanoconfined catalytic system, it would be helpful to supplement experimental evidences about the influence of inorganic anions or organic compounds in the other two control groups.

Response: Accepting the reviewer’s constructive suggestion, we have conducted extra experiments to examine the influences of inorganic anions or dissolved organic matter on the decontamination performances of the other two control groups. The results further confirm a drastically improved environmental robustness of the nanoconfined catalytic system relative to the unconfined control (Supplementary Fig. 32-33). The relevant expressions have been supplemented in the revised manuscript as shown below.

Revision:

Supplementary Figure 32. 4-CP degradation in the presence of different concentration HA. Reaction condition: [catalyst] = $0.25 \text{ g} \cdot \text{L}^{-1}$, [4-CP] = 0.1 mM, [PMS] = 0.4 mM, [HA] = 0-20 mg/L.

Supplementary Figure 33. 4-CP degradation kinetics under different environmentally-relevant ions for two (a) CoNC-MSi2 and (b) CoNC-Si catalysts. Reaction condition: [catalyst] = $0.25 \text{ g} \cdot \text{L}^{-1}$, [PMS] = 0.4 mM, [4-CP] = 0.1 mM, [cations] = [anions] = 5 mM.

“.....The 4-CP degradation performance of the CoNC-MSi1/ PMS system was nearly unaffected by diverse environmentally-ubiquitous inorganic ions (Na^+ , K^+ , Ca^{2+} , Cl, F, CO_3^{2-} , and PO_4^{3-}), natural organic matters (NOM) and in broad pH range (Fig. 6C, Supplementary Fig. 32–34). Here, the steric hindrance effect of the large-molecule NOM might also contributed to the superior NOM resistance of the nanoconfined Co

atoms².” (Page. 11)

4. The figure quality should be improved. For example, the display of Fig. 5 is not complete; what's meaning of the solid line and the dashed line in Figure 4G? The full names of the pollutants should be provided in the figure caption; the font format across all figures should be standardized in the manuscript. A comprehensive refinement of the graphical representations is advised to ensure the consistency and clarity.

Response: Thanks for pointing out these issues. We have carefully checked and updated the figures to address these problems. The meanings of the solid and dashed lines have been added in the related expression of the revised manuscript. The full names of the pollutants were also added in the figure caption. Consistent formats have been adopted in all the figures.

Revision:

“..... Therefore, a ¹O₂-to-ETP pathway transition in Fenton-like catalysis was triggered by the nanoconfinement of Co SACs, contributing considerably to the high decontamination activity and PUE of the catalytic systems. Specifically, a considerable fraction of the generated ¹O₂ in the unconfined system, due to lack of sufficient contact with pollutants, was self-decomposed during diffusion into the bulk solution⁴⁵, resulting in low PUE. In contrast, the nanoconfined system could not only accelerate the PMS activation and pollutant oxidation due to their raised local concentrations, but also provide long-lasting PMS to favor more efficient PMS utilization for pollutant degradation (Fig. 4G).” (Page. 8)*

Figure 6A. Degradation performance of different organic pollutants in CoNC-MSi1/PMS system including 4-CP, Ph (phenol), BPA (bisphenol A), 4-NP (p-nitrophenol), CB (chlorobenzene) and BA (benzoic acid).

5. The English language can be further improved for better clarity and readability. The format of references should be unified following the guidelines.

Response: Thanks. Most part of the main text has been rewritten for better clarity and readability. The reference format has also been standardized as suggested.

Reference

1. Zhang S, Hedtke T, Zhou X, Elimelech M, Kim J-H. Environmental Applications of Engineered Materials with Nanoconfinement. *ACS ES&T Engineering* **1**, 706-724 (2021).
2. Qian J, Gao X, Pan B. Nanoconfinement-Mediated Water Treatment: From Fundamental to Application. *Environmental Science & Technology* **54**, 8509-8526 (2020).
3. Meng C, *et al.* Angstrom-confined catalytic water purification within Co-TiO_x laminar membrane nanochannels. *Nature Communications* **13**, 4010 (2022).
4. Grommet AB, Feller M, Klajn R. Chemical reactivity under nanoconfinement. *Nature Nanotechnology* **15**, 256-271 (2020).
5. Yang Z, Qian J, Yu A, Pan B. Singlet oxygen mediated iron-based Fenton-like catalysis under nanoconfinement. *Proceedings of the National Academy of Sciences* **116**, 6659-6664 (2019).
6. Liu C, *et al.* Nanoconfinement Engineering over Hollow Multi-Shell Structured Copper towards Efficient Electrocatalytical C–C coupling. *Angewandte Chemie International Edition* **61**, e202113498 (2022).
7. Zhou T, *et al.* Nanopore Confinement of Electrocatalysts Optimizing Triple Transport for an Ultrahigh-Power-Density Zinc–Air Fuel Cell with Robust Stability. *Advanced Materials* **32**, 2003251 (2020).
8. Gu C-H, Wang S, Zhang A-Y, Liu C, Jiang J, Yu H-Q. Slow-release synthesis of Cu single-atom catalysts with the optimized geometric structure and density of state distribution for Fenton-like catalysis. *Proceedings of the National Academy of Sciences* **120**, e2311585120 (2023).
9. Zeng Z, *et al.* Single-atom platinum confined by the interlayer nanospace of carbon nitride for efficient photocatalytic hydrogen evolution. *Nano Energy* **69**, 104409 (2020).
10. Jung E, *et al.* Atomic-level tuning of Co–N–C catalyst for high-performance electrochemical H₂O₂ production. *Nature Materials* **19**, 436-442 (2020).
11. Hu J, *et al.* Uncovering Dynamic Edge-Sites in Atomic Co–N–C Electrocatalyst for Selective Hydrogen Peroxide Production. *Angewandte Chemie International Edition* **62**, e202304754 (2023).
12. Chen S, *et al.* Identification of the Highly Active Co–N₄ Coordination Motif for Selective Oxygen Reduction to Hydrogen Peroxide. *Journal of the American Chemical Society* **144**, 14505-14516 (2022).

13. Yu P, *et al.* Co Nanoislands Rooted on Co–N–C Nanosheets as Efficient Oxygen Electrocatalyst for Zn–Air Batteries. *Advanced Materials* **31**, 1901666 (2019).
14. Chen Z, An F, Zhang Y, Liang Z, Liu W, Xing M. Single-atom Mo–Co catalyst with low biotoxicity for sustainable degradation of high-ionization-potential organic pollutants. *Proceedings of the National Academy of Sciences* **120**, e2305933120 (2023).
15. Wang H, *et al.* Sludge-derived biochar as efficient persulfate activators: Sulfurization-induced electronic structure modulation and disparate nonradical mechanisms. *Applied Catalysis B: Environmental* **279**, 119361 (2020).
16. Fei H, *et al.* Atomic cobalt on nitrogen-doped graphene for hydrogen generation. *Nat Commun* **6**, 8668 (2015).
17. Yamaguchi Y, Inouye M. Regulation of growth and death in Escherichia coli by toxin–antitoxin systems. *Nature Reviews Microbiology* **9**, 779–790 (2011).
18. Rajhans G, Barik A, Sen SK, Masanta A, Sahoo NK, Raut S. Mycoremediation and toxicity assessment of textile effluent pertaining to its possible correlation with COD. *Scientific Reports* **11**, 15978 (2021).
19. Miao J, *et al.* Spin-State-Dependent Peroxymonosulfate Activation of Single-Atom M–N Moieties via a Radical-Free Pathway. *ACS Catalysis* **11**, 9569–9577 (2021).
20. Wang A, *et al.* Enhanced and synergistic catalytic activation by photoexcitation driven S–scheme heterojunction hydrogel interface electric field. *Nat Commun* **14**, 6733 (2023).
21. Chen Y, Zhang G, Liu H, Qu J. Confining Free Radicals in Close Vicinity to Contaminants Enables Ultrafast Fenton-like Processes in the Interspacing of MoS₂ Membranes. *Angewandte Chemie International Edition* **58**, 8134–8138 (2019).
22. Yang Z, Qian J, Shan C, Li H, Yin Y, Pan B. Toward Selective Oxidation of Contaminants in Aqueous Systems. *Environmental Science & Technology* **55**, 14494–14514 (2021).
23. Ren W, *et al.* Origins of Electron-Transfer Regime in Persulfate-Based Nonradical Oxidation Processes. *Environmental Science & Technology* **56**, 78–97 (2022).
24. Weng Z, *et al.* Site Engineering of Covalent Organic Frameworks for Regulating Peroxymonosulfate Activation to Generate Singlet Oxygen with 100 % Selectivity. *Angewandte Chemie International Edition* **62**, e202310934 (2023).
25. Xin S, *et al.* Electron Delocalization Realizes Speedy Fenton-Like Catalysis over a High-Loading and Low-Valence Zinc Single-Atom Catalyst. *Advanced Science* **10**, 2304088 (2023).

26. Liu T, *et al.* Water decontamination via nonradical process by nanoconfined Fenton-like catalysts. *Nature Communications* **14**, 2881 (2023).
27. Wang Y, Lin Y, He S, Wu S, Yang C. Singlet oxygen: Properties, generation, detection, and environmental applications. *Journal of Hazardous Materials* **461**, 132538 (2024).
28. Wu Q-Y, Yang Z-W, Wang Z-W, Wang W-L. Oxygen doping of cobalt-single-atom coordination enhances peroxymonosulfate activation and high-valent cobalt–oxo species formation. *Proceedings of the National Academy of Sciences* **120**, e2219923120 (2023).
29. Wu L, *et al.* Oxygen Vacancy-Induced Nonradical Degradation of Organics: Critical Trigger of Oxygen (O₂) in the Fe–Co LDH/Peroxymonosulfate System. *Environmental Science & Technology* **55**, 15400-15411 (2021).
30. Jin C, *et al.* Space-Confined Surface Layer in Superstructured Ni–N–C Catalyst for Enhanced Catalytic Degradation of m-Cresol by PMS Activation. *ACS Applied Materials & Interfaces* **14**, 40834-40840 (2022).
31. Zhang X, *et al.* Nanoconfinement-triggered oligomerization pathway for efficient removal of phenolic pollutants via a Fenton-like reaction. *Nature Communications* **15**, 917 (2024).
32. Wei Y, *et al.* Ultrahigh Peroxymonosulfate Utilization Efficiency over CuO Nanosheets via Heterogeneous Cu(III) Formation and Preferential Electron Transfer during Degradation of Phenols. *Environmental Science & Technology* **56**, 8984-8992 (2022).
33. Yu Z, *et al.* Decoupled oxidation process enabled by atomically dispersed copper electrodes for in-situ chemical water treatment. *Nature Communications* **15**, 1186 (2024).
34. Li X, *et al.* Single Cobalt Atoms Anchored on Porous N-Doped Graphene with Dual Reaction Sites for Efficient Fenton-like Catalysis. *Journal of the American Chemical Society* **140**, 12469-12475 (2018).
35. Sheng X, *et al.* N-doped ordered mesoporous carbons prepared by a two-step nanocasting strategy as highly active and selective electrocatalysts for the reduction of O₂ to H₂O₂. *Applied Catalysis B-Environmental* **176**, 212-224 (2015).
36. Kumar P, *et al.* High-Density Cobalt Single-Atom Catalysts for Enhanced Oxygen Evolution Reaction. *Journal of the American Chemical Society* **145**, 8052-8063 (2023).
37. Yao Y, *et al.* Rational Regulation of Co–N–C Coordination for High-Efficiency Generation of ¹O₂ toward Nearly 100% Selective Degradation of Organic Pollutants. *Environmental Science & Technology* **56**, 8833-8843 (2022).

38. Zang Y, *et al.* Selective CO₂ Electroreduction to Ethanol over a Carbon-Coated CuO_x Catalyst. *Angewandte Chemie International Edition* **61**, e202209629 (2022).
39. Guo Z-Y, *et al.* Crystallinity engineering for overcoming the activity–stability tradeoff of spinel oxide in Fenton-like catalysis. *Proceedings of the National Academy of Sciences* **120**, e2220608120 (2023).
40. Yun E-T, Lee JH, Kim J, Park H-D, Lee J. Identifying the Nonradical Mechanism in the Peroxymonosulfate Activation Process: Singlet Oxygenation Versus Mediated Electron Transfer. *Environmental Science & Technology* **52**, 7032-7042 (2018).
41. Zhao Y, *et al.* Janus electrocatalytic flow-through membrane enables highly selective singlet oxygen production. *Nature Communications* **11**, 6228 (2020).
42. Wang L, Xu H, Jiang N, Pang S, Jiang J, Zhang T. Effective activation of peroxymonosulfate with natural manganese-containing minerals through a nonradical pathway and the application for the removal of bisphenols. *Journal of Hazardous Materials* **417**, 126152 (2021).
43. Zhao Y, *et al.* Fe₃C@nitrogen doped CNT arrays aligned on nitrogen functionalized carbon nanofibers as highly efficient catalysts for the oxygen evolution reaction. *Journal of Materials Chemistry A* **5**, 19672-19679 (2017).
44. Zhang T, Zhu H, Croué J-P. Production of Sulfate Radical from Peroxymonosulfate Induced by a Magnetically Separable CuFe₂O₄ Spinel in Water: Efficiency, Stability, and Mechanism. *Environmental Science & Technology* **47**, 2784-2791 (2013).
45. Yan Y, *et al.* Merits and Limitations of Radical vs. Nonradical Pathways in Persulfate-Based Advanced Oxidation Processes. *Environmental Science & Technology* **57**, 12153-12179 (2023).

REVIEWER COMMENTS

Reviewer #1 (Remarks to the Author):

The authors have addressed most of the raised issues and the quality of this work is significantly improved. However, there are still some issues, which should be addressed to further improve this work.

1. It is interesting to know that the adsorption energy for PMS and 4-CP with the pore size of 7.7 nm is greater than the case with the pore size of 2.7 nm. In other words, the nanoconfinement effect decreases with the decrease of the pore size in this system, which is totally different from the conventional understanding. If that is the case, then theoretically, what is the optimal pore size in terms of nanoconfinement effect for this system. The authors mentioned that the nanopore size can be adjusted, whether the current pore sizes of 7.7 nm and 12.8 nm are (close to) the best in terms of nanoconfinement effect?
2. Fig. 3D-E, These results are the theoretical PMS/4-CP distribution concentration for the distance of 0-2 nm to catalyst surface. However, the mean pore sizes are 7.7 and 12.8 nm. The authors should consider and provide the PMS/4-CP distribution concentration over the mean pore size (0-7.7nm and 0-12.8 nm) rather than 0-2 nm. Moreover, how about the diffusion performance of PMS/4-CP within the pores? -such as MSD results?
3. For the packed-bed reactor, the operation pressure is 10 MPa? That pressure is even higher than the pressure for RO membrane, which is hard and not cost-effective for potential application. Discussion about the further improvement about this point should be supplemented.
4. Fig. 4D-E, it is important to show the experimental time in the manuscript for the double-chamber electrochemical cell results with the detailed results presented in SI.

Reviewer #3 (Remarks to the Author):

The authors have made great effort to revise this manuscript and all of my comments have been addressed properly. Additionally, they also provided serious and reasonable responses to the comments of other reviewers. From the novelty of the manuscript and their attitude towards the revision, I recommend this work for publication.

Reponses to Reviewer #1's Comments

The authors have addressed most of the raised issues and the quality of this work is significantly improved. However, there are still some issues, which should be addressed to further improve this work.

Response: We are grateful for your positive comments and constructive suggestions. Following your valuable suggestions, we have further deepened the discussion to highlight the scientific merit and practical significance of our findings.

1. It is interesting to know that the adsorption energy for PMS and 4-CP with the pore size of 7.7 nm is greater than the case with the pore size of 2.7 nm. In other words, the nanoconfinement effect decreases with the decrease of the pore size in this system, which is totally different from the conventional understanding. If that is the case, then theoretically, what is the optimal pore size in terms of nanoconfinement effect for this system. The authors mentioned that the nanopore size can be adjusted, whether the current pore sizes of 7.7 nm and 12.8 nm are (close to) the best in terms of nanoconfinement effect?

Response: Thanks for your good question. It is true that most previous studies have found an enhanced nanoconfinement effect with decreased size of the confined space (*Nat Commun.* 2022, 13, 4010). However, recent findings suggest that the size-dependence of nanoconfinement effects are actually more complicated than what we have thought, because many influential factors are intertwined to alter the overall catalytic performance. For instance, optimal performance of electrocatalysts for oxygen reduction reaction (ORR) was achieved by applying nanoconfinement with medium-sized nanochannels that favored balanced permeation resistance and ion diffusion (*Adv. Mater.* 2020, 32, 2003251). In another study, the nanoconfined core-shell Ag@Cu catalysts demonstrate the highest activity for CO₂ reduction when medium-sized nanopores of the Cu shells were adopted. (*Nano Lett.* 2022, 22, 6, 2554–2560).

Likewise, the theoretical analysis in our study also suggests similar changing trends

of Fenton-like catalytic activity and pathway for the nanoconfined SAC in response to nanopore size variation. Specifically, the CoNC-MSi1 with medium-sized nanopore (7.7 nm) is supposed to have higher Fenton-like activity and higher fraction of electron transfer process (ETP) pathway than the counterparts with larger (12.8 nm) or smaller pores (2.7 nm, which features weak adsorption of PMS and 4-CP and smaller charge transfer number to hinder ETP) (Fig. 5A–D and Supplementary Fig. 30).

We have tried to fabricate catalyst with smaller-sized nanopores for validation but failed, because 7.7 nm pore size was the limit that could be achieved by the present synthesis method. Therefore, direct experimental evidence regarding the optimal nanopore size for our nanoconfined catalytic system is still lacking. Nevertheless, this does not affect our conclusion that nanoconfinement of SAC can steer the Fenton-like catalysis from singlet oxygen ($^1\text{O}_2$) pathway to electron transfer process (ETP) for 4-CP oxidation.

Indeed, the detailed size-dependent nanoconfinement effects in heterogeneous Fenton-like catalysis are complicated and were not fully clarified yet in this work. We would like to explore further into this interesting issue in our future study. To facilitate a better understanding by the readers, we have rephrased the description of the nanoconfinement effects and highlighted the remaining knowledge gaps.

Revision:

Supplementary Figure 30. (a) Models of PMS and 4-CP binding on nanoconfined catalysts with different pore sizes and the corresponding adsorption energy, (b) Charge density difference for PMS adsorption on the catalysts with 2.7 nm pore size (the light yellow and light blue denote the electron depletion and accumulation, respectively).

Figure 5A–B. Charge density difference for PMS adsorption on catalysts with 7.7 nm and 12.8 nm pore sizes (the light yellow and light blue denote the electron depletion and accumulation, respectively).

“..... Such changes are consistent with the previous reports that the nanoconfinement effects typically became more prominent with reduced size of the confined space¹⁻². However, the size impacts on nanoconfinement may become more complicated when it comes to a level of several nanometer, because of drastic changes in solution chemistry and interfacial molecule interactions that may suppress the catalytic activity instead³⁻⁴. Here, according to our DFT calculation, further downsizing the nanopores of the catalysts from 7.7 nm to 2.7 nm diameter might weaken the binding of PMS and pollutant and result in decreased charge transfer numbers instead (Supplementary Fig. 30 and Table 7). Overall, although the detailed correlation between the nanoconfinement effects and pore size in the SAC catalytic system remain elusive, it seems that an optimal confinement size might exist for maximizing the catalytic performance.” (Page 10)

2. Fig. 3D-E, these results are the theoretical PMS/4-CP distribution concentration for the distance of 0-2 nm to catalyst surface. However, the mean pore sizes are 7.7 and 12.8 nm. The authors should consider and provide the PMS/4-CP distribution concentration over the mean pore size (0-7.7nm and 0-12.8 nm) rather than 0-2 nm. Moreover, how about the diffusion performance of PMS/4-CP within the pores? -such as MSD results?

Response: Following the reviewer's constructive suggestion, we have re-performed the theoretical calculation, and added figures to clearly illustrate the distribution profiles of PMS and 4-CP within the entire pore space for each nanoconfined catalyst (Fig. 3D and Supplementary Fig. 17). The results show that, with increased distance from the pore wall of the CoNC-MSi1, the concentrations of PMS and 4-CP were both sharply raised and peaked at around 0.4-1 nm, and then decreased gradually to 1.0 (i.e., the same as the bulk solution), confirming a drastical surface enrichment of the reactants under nanoconfinement. Similar patterns of reactant concentration distribution were also shown by CoNC-MSi2 but with much weak enrichment effect obtained, in good line with its larger pore size and hence weaker nanoconfinement effects. In contrast, no obvious reactant enrichment was observed for the unconfined control (i.e., CoNC-Si).

In addition, we also conducted MSD analysis to reveal information of reactant diffusion kinetics following the reviewer's suggestion. The MSD-t curves (Fig. 3E) clearly show a much faster PMS diffusion kinetics of the CoNC-MSi1 than CoNC-MSi2. Similar results of 4-CP diffusion were also shown by the two catalysts. Therefore, these results confirm an drastically enhanced local enrichment, surface binding and diffusion of reactants under nanoconfinement, consistent with the previous reports (*J. Am. Chem. Soc.* 2022, 144, 30, 13831–13838; *ACS Nano.* 2020, 14, 12, 16348–16391).

These new figures and the relevant discussion has been added in the manuscript.

Revision:

Figure 3D. Concentration distribution of PMS obtained by molecular dynamic simulations.

Supplementary Figure 17. Concentration distribution of 4-CP obtained by molecular dynamic simulations.

Figure 3E. MSD curves showing reactant diffusion kinetics for different catalysts.

“..... The local enrichment and improved diffusion of PMS and pollutants over the nanoconfined Co SAC should be important reasons for its high specific activity and PUE. The results of molecular dynamic (MD) simulation show a 1.8-fold raised PMS concentration and 3.9-fold raised 4-CP concentration at the proximity of the pore wall for CoNC-MSi1 relative to those of the bulk liquid. In contrast, no obvious concentration lift was observed for the CoNC-Si, while slight concentration rise was shown by CoNC-MSi2 (Fig. 3D, Supplementary Fig. 17). These results confirm an obvious reactant enrichment effect of the nanoconfined catalyst. In addition, the mean square displacement (MSD) curves, which reveals key information of reactant diffusion kinetics⁵, show a much faster reactant diffusion rate for the CoNC-MSi1 than CoNC-MSi2 (Fig. 3E). Therefore, the raised local concentrations and enhanced diffusion of pollutant and PMS should contribute considerably to the high activity of the nanoconfined Co SAC. However, the several times of reactant enrichment for the CoNC-MSi1 could not fully explain its tenfold raised specific activity than that of the

CoNC-Si, indicating that some other factors might also account for the performance improvement under nanoconfinement..”(Page 7)

3. For the packed-bed reactor, the operation pressure is 10 MPa? That pressure is even higher than the pressure for RO membrane, which is hard and not cost-effective for potential application. Discussion about the further improvement about this point should be supplemented.

Response: We appreciate the reviewer’s constructive suggestion, and fully agree that the operating pressure (10 MPa) is too high for practical water treatment applications. In this work, we selected the high-pressure system only aiming to demonstrate that our catalyst can still maintain integrity and high stability even under pressures of up to 10 MPa. In other words, we found that the catalyst was fully adaptable to pressurized operation conditions resembling membrane filtration operation. To validate its feasibility for membrane catalytic decontamination application under low-pressure mode, we loaded CoNC in an inorganic ceramic membrane and evaluated its performance for 4-CP removal under operating pressure of only 0.01 ± 0.003 MPa (Supplementary Fig. 45). The catalytic membrane/PMS system, at water flux of $200 \text{ L m}^{-2} \text{ h}^{-1}$, maintained 100% 4-CP removal efficiency during 5-hour continuous operation. This result demonstrate a great potential of the catalytic system for cost-effective water treatment. The relevant contents have been added in the manuscript.

Revision:

Supplementary Figure 45. The 4-CP removal and operation pressure during catalytic membrane operation. Reaction condition: [PMS] = 0.1 mM, [4-CP] = 0.1 mM, [flux] = 200 L m⁻² h⁻¹, and 4-CP is reached adsorption equilibrium within 2 hours before adding PMS.

“..... Given its superior mechanical stability, environmental robustness and catalytic activity, the CoNC-MSi1 catalyst may be readily incorporated into other water systems to facilitate practical application. Here, we further validated its feasibility for membrane catalytic decontamination application by incorporating the catalyst into an inorganic ceramic membrane. As expected, the catalytic membrane/PMS system, at water flux of 200 L m⁻² h⁻¹ and under an operating pressure of 0.01±0.003 MPa, maintained 100% 4-CP removal efficiency during 5-hour continuous operation (Supplementary Fig. 45). This result demonstrated a great potential of the catalytic system for cost-effective water treatment” (Page 12)

4. Fig. 4D-E, it is important to show the experimental time in the manuscript for the double-chamber electrochemical cell results with the detailed results presented in SI.

Response: Following the reviewer's suggestion, we have added the experimental time information of the double-chamber electrochemical cell assay in the manuscript and Supplementary Fig. 22.

Revision:

Supplementary Figure 22 a. 4-CP degradation performance without and with PMS for 6 hours reaction. Reaction condition: [catalyst] = $0.5 \text{ g} \cdot \text{L}^{-1}$, [PMS] = 0.8 mM, [4-CP] = 0.05 mM.

Note: The dual-chamber galvanic cells are used to assess the ETP process, and the results show 100% 4-CP degradation in two CoNC-MSi/PMS systems for 6-hour operation. In contrast, only 60% 4-CP removal in CoNC-Si/PMS system, consistent with the result of blank carbon paper, suggesting non-ETP-dominated process in CoNC-Si/PMS system. The operation time lasted for 6 hours because of the low current $\sim 14 \text{ nA}$ during the reaction and slower mass transfer limited reaction by salt bridge compared with the batch system.

“..... To elucidate the specific catalyst-pollutant-PMS interactions, we constructed a galvanic cell, where the two electrolyte chambers (containing 4-CP and PMS, respectively) were separated by a catalyst-loaded carbon flake. This system showed distinct current changes upon 4-CP and PMS addition during 6-hour operation, accompanied by synchronous decomposition of the reactants in individual chambers (Fig. 4D–E, Supplementary Fig. 22).”(Page 8).

Reponses to Reviewer #3's Comments

The authors have made great effort to revise this manuscript and all of my comments have been addressed properly. Additionally, they also provided serious and reasonable responses to the comments of other reviewers. From the novelty of the manuscript and their attitude towards the revision, I recommend this work for publication.

Response: We deeply appreciate the reviewer's positive comments and the great help in improving the quality of our manuscript.

Reference

1. Wang, B.; Wang, M.; Fan, Z.; Ma, C.; Xi, S.; Chang, L. Y.; Zhang, M.; Ling, N.; Mi, Z.; Chen, S.; Leow, W. R.; Zhang, J.; Wang, D.; Lum, Y., Nanocurvature-induced field effects enable control over the activity of single-atom electrocatalysts. *Nature Communications* **2024**, *15* (1), 1719.
2. Zhang, S.; Sun, M.; Hedtke, T.; Deshmukh, A.; Zhou, X.; Weon, S.; Elimelech, M.; Kim, J.-H., Mechanism of Heterogeneous Fenton Reaction Kinetics Enhancement under Nanoscale Spatial Confinement. *Environmental Science & Technology* **2020**, *54* (17), 10868-10875.
3. Zhong, Y.; Kong, X.; Song, Z.; Liu, Y.; Peng, L.; Zhang, L.; Luo, X.; Zeng, J.; Geng, Z., Adjusting Local CO Confinement in Porous-Shell Ag@Cu Catalysts for Enhancing C–C Coupling toward CO₂ Electroreduction. *Nano Letters* **2022**, *22* (6), 2554-2560.
4. Meng, C.; Ding, B.; Zhang, S.; Cui, L.; Ostrikov, K. K.; Huang, Z.; Yang, B.; Kim, J.-H.; Zhang, Z., Angstrom-confined catalytic water purification within Co-TiO_x laminar membrane nanochannels. *Nature Communications* **2022**, *13* (1), 4010.
5. Ren, Q.; Gupta, M. K.; Jin, M.; Ding, J.; Wu, J.; Chen, Z.; Lin, S.; Fabelo, O.; Rodríguez-Velamazán, J. A.; Kofu, M.; Nakajima, K.; Wolf, M.; Zhu, F.; Wang, J.; Cheng, Z.; Wang, G.; Tong, X.; Pei, Y.; Delaire, O.; Ma, J., Extreme phonon anharmonicity underpins superionic diffusion and ultralow thermal conductivity in argyrodite Ag₈SnSe₆. *Nature Materials* **2023**, *22* (8), 999-1006.

REVIEWERS' COMMENTS

Reviewer #1 (Remarks to the Author):

For the raised issues, the authors have supplemented proofs to address them. Although the theoretical optimal pore size of the CoNC-MSi for the nanoconfinement effect remains elusive, the authors have clearly pointed out this issue and mentioned the further investigations in the future. As such, the current version can be accepted.